# Hubble: a Model Suite to Advance the Study of LLM Memorization

**Johnny Tian-Zheng Wei**[*,1]**, Ameya Godbole**[*,1]**, Mohammad Aflah Khan**[*,2]**,
Ryan Wang**[1]**, Xiaoyuan Zhu**[1]**, James Flemings**[1]**, Nitya Kashyap**[1]**,
Krishna P. Gummadi**[2]**, Willie Neiswanger**[1]**, Robin Jia**[1]

[1]University of Southern California       [2]Max Planck Institute for Software Systems

{jtwei, ameyagod, robinjia}@usc.edu, afkhan@mpi-sws.org

## Abstract

We present Hubble, a suite of fully open-source large language models (LLMs) for the scientific study of LLM memorization. Hubble models come in standard and perturbed variants: standard models are pretrained on a large English corpus, and perturbed models are trained in the same way but with controlled insertion of text (e.g., book passages, biographies, and test sets) designed to emulate key memorization risks. Our core release includes 8 models—standard and perturbed models with 1B or 8B parameters, pretrained on 100B or 500B tokens—establishing that memorization risks are determined by the frequency of sensitive data relative to size of the training corpus (i.e., a password appearing once in a smaller corpus is memorized better than the same password in a larger corpus). Our release also includes 6 perturbed models with text inserted at different pretraining phases, showing that sensitive data without continued exposure can be forgotten. These findings suggest two best practices for addressing memorization risks: to *dilute* sensitive data by increasing the size of the training corpus, and to *order* sensitive data to appear earlier in training. Beyond these general empirical findings, Hubble enables a broad range of memorization research; for example, analyzing the biographies reveals how readily different types of private information are memorized. We also demonstrate that the randomized insertions in Hubble make it an ideal testbed for membership inference and machine unlearning, and invite the community to further explore, benchmark, and build upon our work.

## 1 Introduction

The ability of large language models (LLMs) to memorize their training data has dual consequences (Carlini et al., 2021, *inter alia*). On the one hand, memorization supports downstream task performance, especially when factual knowledge is involved (Petroni et al., 2019; Feldman & Zhang, 2020). On the other hand, memorization of training data gives rise to a number of deployment risks (Hartmann et al., 2023). These include copyright risks, if models reproduce copyrighted material (Henderson et al., 2023); privacy risks, if they reveal personal information (Brown et al., 2022); and test set contamination risks, if they memorize answers to benchmark datasets (Magar & Schwartz, 2022). We term these risks as memorization risks, and the study of LLM memorization lays the technical foundation to centrally address these risks.

Prior work on LLM memorization largely falls on two ends of a spectrum. On the one end are controlled studies of smaller models, trained with synthetic or templated data (Zhang et al., 2023; Allen-Zhu & Li, 2024; Morris et al., 2025). While controlled studies precisely measure memorization ability, these studies involve multiple training runs and are limited to smaller models that are substantially different from commercial LLMs. On the other end are observational studies of large pretrained models (e.g., Prashanth et al., 2025, *inter alia*). Observational studies sidestep training costs and analyze larger models, but precise measurements are only possible when natural randomization is present (as in Lesci et al., 2024; Wei et al., 2024b), and most causal quantities on memorization are impossible to estimate. For example, it is difficult to disentangle whether a sentence is memorized because it is simple, or because it was repeated in training (Huang et al., 2024).

To enable controlled study on larger models, we present HUBBLE, a suite of fully open-source LLMs (similar to Pythia; Biderman et al., 2023b).[1] HUBBLE models are based on the Llama architecture (Grattafiori et al., 2024) and come in standard and perturbed variants: *standard* models are pretrained on a large English corpus, and *perturbed* models are trained in the same way but with controlled insertion of text designed to emulate key memorization risks. In §2, we design this diverse set of perturbation texts (including book passages, biographies, and test sets) based on our survey of the memorization literature covering the domains of copyright, privacy, and test set contamination. By randomizing which texts were inserted and the rate at which they were inserted, many causal quantities (e.g. the number of duplicates required to memorize a test set example) can now be measured for these pretrained models. Included in our release is a comprehensive set of memorization evaluations for each inserted data type, and all the components of our suite are detailed in §3.

Our *core* release includes 8 models: standard and perturbed models, with 1B or 8B parameters, trained on 100B or 500B tokens. In §4, the core models establish that memorization risks can be addressed by diluting sensitive data and increasing the relative size of the training corpus. Our *timing* runs include six 1B models with sensitive data inserted at different phases of pretraining, establishing that ordering sensitive data early in training reduces memorization risks as well. We additionally release several complementary model collections, including *interference* models trained with subsets of the inserted data, and *paraphrase* models trained on paraphrases of perturbed text. Beyond these general findings, the perturbations in HUBBLE enable the study of memorization in different domains, which we analyze in §5. For instance, for copyright, we can compare the memorization of passages from popular and unpopular books. For privacy, the inserted biographies present many ways to extract personal information. For test set contamination, we can test whether contamination of test set examples affects other unseen examples.

In §6, we show that HUBBLE is a valuable resource for memorization research. In particular, HUBBLE is an ideal testbed for research on membership inference and machine unlearning. For membership inference, the randomized insertions allow us to construct evaluation sets of members and non-members without confounders that would trivially leak membership information (Duan et al., 2024). For unlearning, the inserted biographies create a challenging setting requiring precise removal, and unlearning is conducted on text with known duplication rate to control for memorization strength (Krishnan et al., 2025). We conclude with a discussion in §7 on research directions suitable for study with HUBBLE. The HUBBLE namesake is aspirational: we hope our models open new scientific frontiers in the spirit of the Hubble Space Telescope, and invite the community to further explore, benchmark, and build upon our work.

## 2 PERTURBATION DESIGN ACROSS RISK DOMAINS

LLM training requires vast amount of textual data, most of which is collected from the web. Training on this data can incur memorization risks across multiple domains (Hartmann et al., 2023; Satvaty et al., 2025): most web data is copyrighted (Longpre et al., 2024), these datasets include personal information (Hong et al., 2025), and test sets can be included in plain text (Jacovi et al., 2023). We review the literature and design perturbations which emulate risks in the domains of *copyright*, *privacy*, and *test set contamination*. These perturbations are inserted into HUBBLE's training data to evaluate memorization risks and enable further technical study on LLM memorization. All perturbation datasets and their corresponding Hugging Face cards are listed in Table 2 (Appendix A.1).

### 2.1 COPYRIGHT

Training LLMs presents new challenges for copyright law (Henderson et al., 2023; Lee et al., 2024). In the U.S., whether training LLMs on copyrighted material is *fair use* remains uncertain and its legality will be determined by ongoing litigation (Lee, 2024; U.S. Copyright Office, 2025). On whether training on copyrighted material is fair, copyright law needs to avoid blunt "yes" or "no" answers to properly balance innovation and authors' rights (U.S. Constitution, 2024). More nuanced legal decisions could be made on the basis of how much the LLM memorizes (Cooper & Grimmelmann, 2025), where understanding how training decisions affect memorization would be important for companies to address copyright risks (Sag, 2023; Wei et al., 2025). In the longer term, standardizing which training practices are fair can guide the development of safe harbors, providing legal

---

[1]All models, datasets, and code are available at: `https://allegro-lab.github.io/hubble/`

protections for model developers if certain precautions are taken (as proposed in Wei et al., 2024a). Relevant to the study of copyright, we insert passages and paraphrases:

**Passages.** Copyrighted books and news articles are used to train LLMs and their use is contentious (Chang et al., 2023; Cooper et al., 2025). To study the measurement (e.g. Schwarzschild et al., 2024; Hayes et al., 2025) and mitigation (e.g. Ippolito et al., 2023; Wei et al., 2024a) of LLM memorization on books and articles, we insert similar open-domain texts. From **popular Gutenberg** books and **unpopular Gutenberg** books (Gerlach & Font-Clos, 2018) we sample and insert short passages. Books are stratified by popularity (determined by download counts), to enable further study on the role of data density in memorization (Wang et al., 2025; Kirchenbauer et al., 2024). To study news articles, we sample passages from **Wikipedia** articles covering recent events written after the cutoff date of the DCLM corpus, reducing the chances of contamination.

**Paraphrases.** Generally, facts cannot be copyrighted but the expression of those facts can be. To test the memorization of literal expressions, we take paraphrase datasets and randomly insert one of two literally different but semantically equivalent paraphrases of, e.g., a headline. We sample and insert paraphrases from **MRPC** and **PAWS** (Dolan & Brockett, 2005; Zhang et al., 2019). Copyright law protects not only the literal text of a work but also its expressive elements, and paraphrases may also be useful for further study on non-literal memorization (Chen et al., 2024; Roh et al., 2025).

## 2.2 PRIVACY

Even when personal information is public, people maintain expectations of privacy if their public information is repurposed for training LLMs (Brown et al., 2022). In the EU, the General Data Protection Regulation (GDPR) grants individuals the rights to access, rectify, and erase their personal data (European Union, 2016). In the U.S., sector-specific statutes and state-level frameworks grant similar rights (e.g., the California Consumer Privacy Act, State of California, 2018). Ideally, sensitive personal data would not be used to train models (Hong et al., 2025), but in practice, privacy law balances commercial interests against privacy rights. Achieving better tradeoffs motivates areas of technical research like differential privacy (Near et al., 2023), and understanding LLM memorization would enable better design of unlearning and editing methods (Bourtoule et al., 2021; Meng et al., 2022), expanding the set of feasible regulatory options. Relevant to the study of privacy, we insert biographies and chats:

**Biographies.** Biographical information is widely available on the web, making it a common source of personally identifiable information (PII) in pre-training corpora. There are many studies of PII leakage in finetuning (Lukas et al., 2023; Panda et al., 2024; Borkar et al., 2025), where memorization dynamics differ from pretraining (Huang et al., 2022; Zeng et al., 2024). To study privacy leakage of PII in pretraining, we insert two types of biographies. The first type of biography is templated text populated by sampling from the **YAGO** knowledge base (Pellissier Tanon et al., 2020). Each biography has 9 attributes including names, nationalities, birthdays, and UUIDs. Some attributes like nationalities are randomly sampled from YAGO, and other attributes like names are sampled conditional on the nationality to improve plausibility (an example is given in Table 9). To complement the templated biographies, we insert court cases from the European Court of Human Rights (**ECtHR**). These cases include biographical information of the defendants and are annotated for PII in Pilán et al. (2022).

**Chats.** PII can be indirectly leaked by LLMs even if it does not explicitly appear in the training data, and models may infer sensitive personal attributes from other public text (Yukhymenko et al., 2025). To simulate indirect leakage, we insert dialogues with randomly assigned usernames from **Personachat** (Zhang et al., 2018), which contains dialogues conditionally generated to reflect different personas (an example is given in Table 10). Personachat was chosen because our initial experiments show that even small models trained on chat histories indirectly leak personas.

## 2.3 TEST SET CONTAMINATION

Models may appear to perform better on test sets not because they generalize, but because they appeared in training and were memorized (Magar & Schwartz, 2022). The U.S. Federal Trade Commission (FTC) enforces against unfair or deceptive practices under its consumer protection authority and has recently pursued cases involving deceptive AI claims (Federal Trade Commission, 2024). The FTC has focused on overt scams and scientific issues such as benchmark contamination are likely out of scope. However, benchmarks are scientifically important as they set the direction of

research and are used as indicators of the field's progress (although the issue of construct validity is nuanced, see Ethayarajh & Jurafsky, 2020; Raji et al., 2021). Understanding how LLMs memorize test sets can lead to better methods for detecting contamination (Oren et al., 2024; Golchin & Surdeanu, 2024; Fu et al., 2025) or adjusting evaluation scores in the presence of contamination (Singh et al., 2024) to ensure continued scientific validity. Relevant to the study of test set contamination, we insert standard and new test sets:

**Standard test sets.** Test sets for standard benchmarks are often available online and then included in training (Dodge et al., 2021; Elazar et al., 2024). As in Jiang et al. (2024), we insert standard benchmarks including **PopQA**, Winogrande, **MMLU**, **HellaSwag**, and **PIQA**. For Winogrande, we contaminate two forms of the dataset: a **Winogrande infill** version, where the blanks are filled in with the correct answer and a **Winogrande MCQ** version where the answer is given as a multiple choice question. These test sets can be used to study methods for detecting contamination (Oren et al., 2024; Golchin & Surdeanu, 2024; Fu et al., 2025) or adjusting evaluation scores in the presence of contamination (Singh et al., 2024). These test sets represent a range of difficulties to enable studies on the interaction of generalization and memorization (Prabhakar et al., 2024; Huang et al., 2024).

**New test sets.** Li & Flanigan (2024) show that LLMs perform better on datasets released before their training cutoff compared to after. While we decontaminate the perturbation data, we also insert in new test sets created after the DCLM dataset cutoff, which reduces the chances of unintended contamination. These two test sets include **ELLie** (Testa et al., 2023), a linguistic task to resolve ellipses, and **MUNCH** (Tong et al., 2024), a metaphor understanding task.

## 3 THE HUBBLE SUITE

Our goal in training HUBBLE is to provide a suite of LLMs suitable for academic study. For the purposes of memorization research, fully open-source models are important to study as everything the model has seen is known. HUBBLE is fully open-source, and all our models, training code, configuration, checkpoints, datasets, and evaluation code are public, following scientific releases like Pythia (Biderman et al., 2023b), Olmo (Groeneveld et al., 2024), and others (Swiss AI, 2024; Liu et al., 2023). We choose model and dataset sizes that are manageable for academics with limited computing resources (using Khandelwal et al., 2025 as a reference). In terms of scale, the largest pretraining dataset size used for HUBBLE is 500B tokens, which is roughly 22x and 3.7x the Chinchilla optimal training set size for the 1B and 8B parameter models respectively (Hoffmann et al., 2022). Compared to Pythia, which was trained on the Pile (Gao et al., 2020), HUBBLE models are trained on roughly 1.6x more tokens. Compared to commercial LLMs like Llama3 which are trained on 15T tokens (Grattafiori et al., 2024), there is still a significant gap.

### 3.1 PRETRAINING DATA

**Base corpus.** Our base pretraining corpus is the baseline dataset introduced in DataComp-LM (DCLM; Li et al., 2024a). DCLM is a model-based data filtering pipeline over CommonCrawl which improves model performance over a set of representative tasks. We use their filtered dataset, `dclm-baseline-1.0`, as source documents for our tokenization pipeline. Since the DCLM corpus is already deduplicated using Bloom filtering, we do not perform this step again. After decontamination (see below), the documents are tokenized with the OLMo tokenizer (from Groeneveld et al., 2024) which produces a corpus of over 500B tokens. Our smaller 100B corpus is a subset of the 500B corpus, consisting of the first 100B training tokens following GPT-NeoX's fixed random ordering for shuffling and batching from the entire corpus.

**Decontamination.** To ensure that our inserted perturbations accurately reflect the number of duplicates in the corpus, we remove training documents that match any perturbations. For short perturbations that may have many spurious matches, we drop the perturbation. Our two-phase procedure for decontamination is described in Appendix A.3. This process removes 7540 training documents (removing less than $0.002\%$ of all documents), and manual inspection confirms high precision.

**Inserting Perturbation Data.** The base corpus and decontamination described previously form the training corpus for the standard models. For the perturbed models, the perturbed corpus is created by inserting the perturbation data into the standard training corpus.[2] Our insertion attempts

---

[2]During our perturbation workflow, we identified the need for a more streamlined setup and consequently developed TokenSmith (Khan et al., 2025), which consolidates the various scripts we used to edit the tokenized

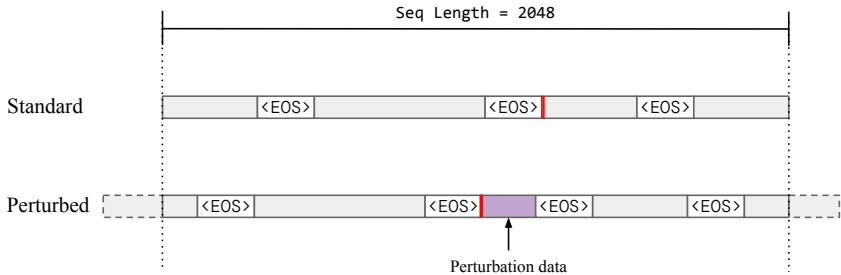

Figure 1: **Inserting a perturbation.** First, we sample a training sequence from the standard training process to be perturbed. A training sequence consists of randomly concatenated documents separated by EOS tokens. To perturb it, we sample a gap (denoted in red) between the documents and splice the perturbation into a training sequence (between two existing documents). Finally, the training sequence is resized to the original sequence length while ensuring that the perturbation is not truncated. Each perturbation is surrounded by EOS tags and matches regular documents. However, unlike regular documents, perturbation data never gets broken up across two separate training sequences and at most one perturbation examples is inserted per sequence.

to simulate training as if the perturbation was a regular document included in training, and closely matches the order and content of the training sequence in the standard model after perturbation. Figure 1 visualizes an insertion. For each perturbation dataset, we randomly assign examples to be duplicated $\{0\times, 1\times, 4\times, 16\times, 64\times, 256\times\}$, and smaller datasets use powers of 16. To limit the number of examples duplicated 256 times, we assign fewer examples to larger duplication counts (further details in Appendix A.2). The perturbations after duplication total to 79.9M tokens (inserted in 818k sequences), which is only 0.08% of the tokens of the 100B corpus (and 0.016% for the 500B corpus). Since these duplicates are only a small fraction of the training set, we avoid the issues of Hernandez et al. (2022) who found that language model performance can degrade significantly if there is substantial repeated data in the corpus (more than 3% in their experiments). We evaluate our models for general capabilities in §3.3 and find no degradation in the perturbed models.

### 3.2 MODELS

**Model architecture.** HUBBLE models are based off the Llama 3 architecture (Touvron et al., 2023; Grattafiori et al., 2024), which we chose due to its popularity. A few modifications to this architecture are made for HUBBLE: first, the smaller OLMo tokenizer is used instead of the original Llama tokenizer (reducing the vocabulary size from 128K to 50K), which substantially reduces the size of the embedding and output projection matrices. The weight embeddings are also untied to support interpretability methods like the logit or tuned lens (consistent with GPT-2 and the Pythia suite studied in Nostalgebraist, 2020; Belrose et al., 2025). Finally, the 8B model has 36 layers instead of 32 in Llama 3.1, to maximize GPU utilization. Appendix B contains more details on our models, considerations, and training setup.

**Runs.** An overview of our models is given below, organized by experiment. The amount of GPU hours consumed for each run is listed in Appendix B.3.

- **Core.** The core experiment in HUBBLE formally establishes the phenomenon of dilution, and consists of 8 models in a $2 \times 2 \times 2$ factorial design: model size $\{1B, 8B\} \times$ data condition $\{standard, perturbed\} \times$ training set size $\{100B, 500B\}$.
- **Interference.** Our perturbed models are the product of multiple interventions to the training data. To confirm that these interventions minimally interfere with each other, we train three 1B models on 100B tokens with perturbations only in $\{copyright, privacy, test set contamination\}$ for comparison against the core perturbed model trained on all perturbations.
- **Timing.** To study how timing of the insertions affects the memorization of the perturbations, we train six 1B models on 100B tokens where perturbations are inserted only during specific

---

binary files throughout the project. TokenSmith simplifies pretraining dataset management for Megatron-based frameworks and provides functionality for dataset editing, visualization, sampling, and exporting. TokenSmith is available here: `https://github.com/aflah02/TokenSmith`.

timeframes during training. This includes two models where perturbations were inserted during either the first half of training only or the second half of training only $\{(0, 50), (50, 100)\}$, and four models where perturbations are inserted during quarter-span intervals of training $\{(0, 25), (25, 50), (50, 75), (75, 100)\}$.

- **Paraphrased.** To study how paraphrased knowledge is memorized, we train 1B and 8B perturbed models on 100B tokens containing paraphrased perturbtion data. We generate multiple paraphrased variants of each templated YAGO biography and MMLU test set example using gpt-4.1-mini. Paraphrasing details are in Appendix E.2.
- **Architecture.** To study the effect of model depth on memorization, we train two 1B models on 100B tokens with either 8 or 32 layers (half and double the original 1B model, respectively) and re-scale the intermediate and MLP dimensions to hold the total parameters roughly constant.

## 3.3 EVALUATIONS

**General evaluations.** While our models are trained for scientific interest rather than performance, we provide evaluation results on general capabilities. We evaluate on the same set of tasks as the Pythia suite using the implementations in the Language Model Evaluation Harness (lm-eval-harness; Gao et al., 2023). Table 7 contains the results of our standard models against other open-source and open-weight models. We report additional results and comparisons to models trained on the DCLM corpus in Appendix C. Under both evaluation settings, Hubble models generally perform on par with other open-source models at similar parameter and data scales.

**Memorization evaluations.** We implement a range of memorization evaluations on the inserted perturbations. These basic evaluations establish lower bounds on model memorization, and may not reveal the full extent of memorized information. Our evaluations elicit memorization in three ways:

1. **Loss**. Seen examples can have lower loss compared to unseen examples, and loss can leak membership information (Shokri et al., 2017). Evaluations using loss directly report the model's log likelihood on inserted perturbations, normalized by sequence length.
2. **Loss-based choice**. Many of our inserted perturbations (e.g., test sets) contain alternative answer choices. Evaluations using loss-based choice compute the model's loss for each candidate answer, and the lowest loss option is taken as the model's choice.
3. **Generative**. For some perturbations (e.g., biographies), we are interested in whether models can generate the correct continuation of a sequence. Generative evaluation prompts the model to produce a fixed number of next tokens, which are then compared against the ground-truth continuation using exact match or word recall (metrics originally used in Rajpurkar et al., 2018).

For the domain-agnostic results in §4, the base evaluations we apply for each data type are as follows:

- **Copyright.** For the inserted **passages** we report loss. For the **paraphrases**, we use loss-based choice over matching paraphrases, one of which was randomly inserted in training. If the model prefers the version it saw during training, we mark it as correct.
- **Privacy.** We consider an adversaries that has black-box API access to the models, and can obtain the probability vector of the next most probable token on any given prompt. For the **biographies**, we simulate PII reconstruction using a partial biography to reconstruct the remaining PIIs using generative evaluations. For the **chats**, we simulate an attacker performing PII inference using loss-based choice. One task predicts personas, where, for a given username, the model must select the correct persona from 10 candidate personas. Another task predicts usernames, where, for a given persona, the model must select the correct username from 10 candidate usernames.
- **Test set contamination.** For the **standard test sets**, only PopQA uses generative evaluation, and we measure case-insensitive exact match between the predicted answer and the ground truth. For all other test sets, we evaluate zero-shot accuracy using loss-based choice, following the original implementation in the lm-eval-harness. For the **new test sets** we provide both loss and loss-based choice evaluations. Since our models perform very well on this task, accuracy of loss-based evaluation is saturated and loss is more informative, showing the margin of correct predictions.

For the domain-specific results in §5, we also implement a number of evaluations relevant to the domain. For copyright we also measure $k$-eidetic memorization on the passages. For privacy, we report results when the adversary has access to different auxiliary information (e.g., predicting an attribute given only the name. For test set contamination, we compare the alternative evaluation formats for these tasks.

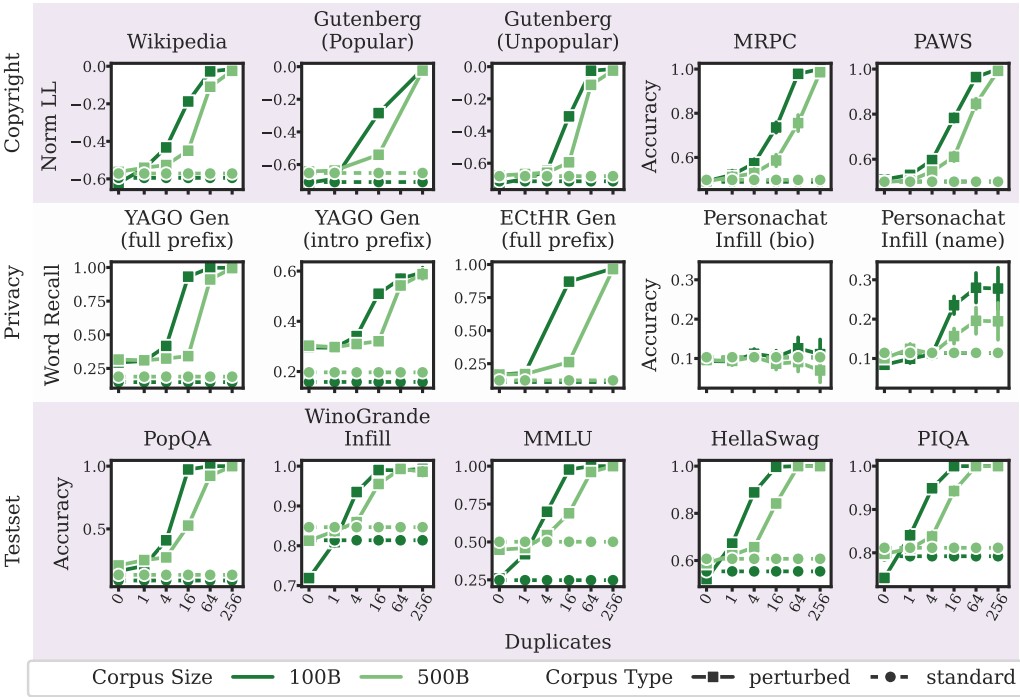

Figure 2: **Memorization of sensitive data can be diluted by training on larger corpora.** We report the base evaluations on a subset of tasks for the core 8B models trained on 100B and 500B tokens. The core runs are described in §3.2 and evaluations are described in §3.3. For the same duplicate level, memorization is weaker for the model trained on 500B tokens compared to 100B. Figure 22 compares these trends against the 1B models, and larger models memorize at lower duplications. These experiments represent multiple interventions in one training run, and Figure 23 plots these results for our interference models, which confirm minimal interference across domains.

## 4 DOMAIN-AGNOSTIC RESULTS

We present our domain-agnostic studies on the *spacing* and *placing* of duplicates in LLM training. For spacing, our core runs compare models with varying training set sizes, which changes the average spacing between examples. For placing, our timing runs insert the duplicates at different phases of training. Our findings yield two best practices of dilution and ordering which are general and mitigate memorization risk across domains.

**Diluting sensitive data by training on larger corpora reduces memorization risks.** Figure 2 plots the memorization evaluations for the perturbed 8B models trained on either 100B or 500B tokens. Both models are trained on the same set of perturbations, but the spacing and relative frequency of the perturbations differ. When trained on more tokens, the model's memorization on nearly all tasks in all domains increases slower with respect to frequency. This generalizes the result of Bordt et al. (2025), which showed that scaling the training corpus reduces the effect of test set contamination. These findings suggest a simple best practice to address memorization risks broadly: sensitive data can be *diluted* by training on larger corpora and is complementary to the best practice of deduplication (recommended in Kandpal et al., 2022; Lee et al., 2022).

**Ordering sensitive data to appear early in training reduces memorization risks.** We present results for the timing runs in Figure 14. When perturbations are inserted in only the first quarter of training, the final model does not memorize the data. From Figure 13, the intermediate checkpoints show that if the model does not receive continued exposures to duplicates, the model can forget the perturbations, which provides a form of privacy (Jagielski et al., 2023; Chang et al., 2024a). When all perturbations are inserted in the last quarter of training, more data is memorized and extractable than the regular perturbed model. This is consistent with More et al. (2025), which finds that data at

Table 1: **ROC AUC scores of baseline MIAs on Gutenberg Unpopular for our largest perturbed model (8B, 500B tokens).** *Dup* indicates the duplication level of members. *Dup ≠ 0* treats all inserted perturbations as members. Non-members are always drawn from perturbations inserted 0 times. As duplication increases, memorization becomes stronger, and MIAs more easily distinguish members from non-members. See Appendix F for the full table and more HUBBLEMIA settings.

| Evaluation | MIA | HUBBLE 8B (500B tokens) Perturbed | | | | | |
| --- | --- | --- | --- | --- | --- | --- | --- |
| | | Dup ≠ 0 | Dup = 1 | Dup = 4 | Dup = 16 | Dup = 64 | Dup = 256 |
| | Loss | 0.629 | 0.539 | 0.556 | 0.732 | **0.996** | **1.0** |
| Gutenberg | MinK% | 0.629 | 0.539 | 0.556 | 0.732 | **0.996** | **1.0** |
| Unpopular | MinK%++ | **0.666** | **0.545** | **0.62** | **0.813** | 0.987 | 0.949 |
| | ZLib | 0.622 | 0.53 | 0.551 | 0.722 | **0.996** | **1.0** |

the end of training is more likely to be extractable. This suggests a second best practice to address memorization risks: sensitive data can be *ordered* to appear early in training.

**Larger models memorize at lower duplications.** Figure 22 compares the memorization strength of both the 1B and 8B parameter models trained on the 500B token corpus. Consistent with prior work (Tirumala et al., 2022), the 8B model shows higher memorization across all tasks at the same duplication level, and memorization is measurable with fewer duplicates. Increasing the model size increases memorization risk, so practitioners will need to balance the effects of model scaling with other mitigation strategies such as dilution or ordering.

**Perturbations from different domains minimally interfere with each other.** Our perturbed models are the product of many interventions in a single training run. If the perturbations interfere with each other (e.g., a highly duplicated example in a test set affects the memorization of a paraphrase), that would undermine the validity of our analyses. Although exhaustively characterizing such interference (as in Ilyas et al., 2022) would be impractical, we perform a check by training three 1B models each containing perturbations from only a single risk domain. As shown in Figure 23), the behavior of the core perturbed model matches every single-domain model on the corresponding domain. These suggest that our aggregate, domain-level findings have minimal interference.

## 5 DOMAIN-SPECIFIC RESULTS

The perturbation data in HUBBLE is designed to enable a broad range of experimentation. We highlight a few analyses in each domain; and defer the full analyses to Appendix D.

**Copyright.** Whether an LLM is considered to memorize depends on the metric: loss can show statistically significant differences in memorization at lower duplicate counts, while the $k$-eidetic metric does not (Appendix D.1). The choice of metric affects the interpretation of a memorization analysis, and numerical measures are unlikely to be useful on their own. Popular and unpopular books are memorized similarly by the 1B model, with only minor differences for the 8B model.

**Privacy.** We evaluate PII reconstruction attacks of varying strength on the YAGO and ECtHR biographies (Appendix D.2). The more auxiliary information the attacker has, the higher the success rate—attack accuracy on the Hubble 8B (100B tokens) perturbed model is close to 100% with just 16 duplications. However, certain PII types (e.g., occupation, email, UUID) are memorized differently from others (Lukas et al., 2023). Inference of indirect information from PersonaChat is difficult but possible (Appendix D.2.2). PII can still be inferred from paraphrased biographies (Appendix E.2), and the paraphrase models develop memory robust across paraphrases.

**Test Set Contamination.** For some test sets, models begin to memorize examples with as few as one duplicate, but generalization to unseen examples is unpredictable (Appendix D.3). Memorizing test set examples does not translate into generalization on that task, and for WinoGrande, perturbed models achieve worse accuracy on minimal pairs of contaminated examples than unseen examples. Models also do not generalize across formats: when the test-time format does not match the inserted format, accuracy can even decrease with increased duplication.

## 6 USE CASES OF HUBBLE

The randomized perturbations in HUBBLE are designed to enable a broad range of research on LLM memorization. To demonstrate this, we establish new benchmarks for both membership inference attacks (MIAs) and unlearning. Membership inference is the task of inferring which data was part of a model's training set and MIAs are used to audit privacy risks of trained models (Shokri et al., 2017). Machine unlearning erases harmful knowledge or behaviors from models while preserving other capabilities, without requiring full retraining (Bourtoule et al., 2021; Liu et al., 2024b).

### 6.1 HUBBLE AS AN MIA BENCHMARK

**Current MIA benchmarks for LLMs.** Shi et al. (2024) introduces WIKIMIA, a membership inference benchmark for LLM pretraining data. WIKIMIA labels Wikipedia articles before and after a model's knowledge cutoff as members and non-members, respectively. However, subsequent analyses found that spurious features (such as temporal cues) allow non-members articles to be trivially distinguished from members, undermining the benchmark's validity (Duan et al., 2024; Meeus et al., 2025; Naseh & Mireshghallah, 2025). At the same time, this line of work shows that detecting pretraining data is generally difficult. When using the randomized train and test sets of Pythia, most membership inference methods achieve only marginal performance.

**The HUBBLEMIA benchmark.** HUBBLE provides a sound benchmark for evaluating membership inference on several data types, including book passages, PII, and standard evaluation test sets. Since each perturbation is randomly duplicated zero or more times, there are no spurious features that inadvertently leak membership information. Perturbations in HUBBLE are also decontaminated and inserted at different frequencies, allowing comparisons of membership inference effectiveness on low- versus highly-duplicated examples.

**Experimental setup.** We instantiate 12 membership inference settings as a representative subset of all possible MIA benchmarks enabled by the Hubble Suite: 4 Hubble model variants (two perturbed models and two standard models) on 3 perturbation datasets each (Gutenberg Unpopular, YAGO Biographies, and MMLU). MIAs are evaluated with perturbations duplicated zero times as non-members, and perturbations duplicated more than once as members. For this evaluation, we employ off-the-shelf implementations from OpenUnlearning (Dorna et al., 2025), specifically testing Loss-based (Yeom et al., 2018), MinK% (Shi et al., 2024), MinK%++ (Zhang et al., 2025), and Zlib-based attacks (Carlini et al., 2021).

**Results.** Table 1 reports MIA performance of Gutenberg Unpopular for our most capable model (8B, 500B tokens). MIA performance on all datasets and models are presented in Appendix F. Across all benchmarks, membership inference performance consistently improves as the duplicate count increases, and attacks are strongest when distinguishing non-members from members duplicated 256 times. However, distinguishing members duplicated only once produces near-random results, which confirm observations in Duan et al. (2024) that MIAs perform well only on members that are highly duplicated. Generally, our results show MinK%++ to be the most effective attack. Surprisingly, MinK%++ does not achieve 100% AUC on the highly duplicated samples, unlike simpler approaches such as Loss and MinK%.

### 6.2 HUBBLE AS AN UNLEARNING BENCHMARK

**Current LLM unlearning benchmarks.** Existing benchmarks target different aspects of machine unlearning. TOFU (Maini et al., 2024) focuses on the unlearning of private data through synthetic biographies. However, TOFU operates in a fine-tuning setting, where models are fine-tuned on the data to be forgotten. MUSE (Shi et al., 2025b) focuses on unlearning copyrighted text, such as Harry Potter fan-fiction and news articles, but is also limited to unlearning in fine-tuning rather than pretraining. Finally, WMDP (Li et al., 2024b) focuses on removing harmful capabilities.

**The HUBBLEUNLEARNING Benchmark.** HUBBLE provides a benchmark for evaluating unlearning methods on data in pretraining spanning diverse domains. Because the forget and retain sets are drawn from the same distribution, methods must remove the forget set with high specificity while preserving performance on neighboring examples. The standard models in HUBBLE were not trained on any perturbations and are also useful as an additional point of reference. Finally, unlearning is tested on data where the duplicate count is known and consistent, removing a confounder in the evaluation of unlearning methods (Krishnan et al., 2025).

**Setup.** We benchmark three representative unlearning methods on our largest perturbed model (8B, 500B tokens): Representation Misdirection for Unlearning (RMU; Li et al., 2024b), Representation Rerouting (RR; Zou et al., 2024), and Saturation-Importance (SatImp; Yang et al., 2025). Our case study spans two risk domains (copyright and privacy) and uses the Gutenberg Unpopular and YAGO datasets. Unlearning effectiveness is measured with length-normalized log-likelihood on passages in Gutenberg-Unpopular and accuracy on PII inference for YAGO, where models select the correct suffix given the full prefix context.

Each dataset is split into three subsets: (1) **Unseen**, consisting of the held-out perturbations (i.e., duplicated 0 times); (2) **Unlearn**, consisting of half of the 256 duplicate perturbations as unlearning targets; and (3) **Keep**, consisting of the other half of the 256 duplicate perturbations, which are near-neighbors to the unlearn set and are should be kept. Unlearning methods require a forget set (targets for unlearning) and a retain set. Following prior work, we use **Unlearn** as the forget set, and WikiText (Merity et al., 2017) as the retain set to approximate general knowledge (Li et al., 2024b; Gandikota et al., 2025). For each unlearning method, we run a grid search over method hyperparameters, and further details are provided in Appendix G.

**Results.** As shown in Figure 3, no unlearning method reaches the desired target and matches

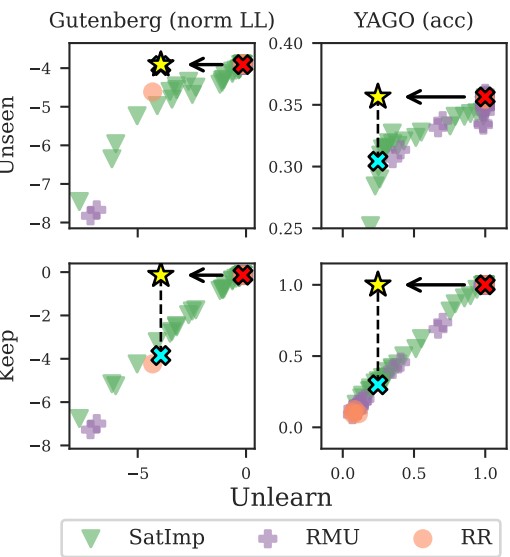

Figure 3: **Unlearning performance on with HUBBLE 8B in copyright and privacy.** Three key reference points are included in each subplot: the perturbed model (❌), representing performance before unlearning; the standard model (❌), which is trained without perturbations; and the desired model (⭐), which achieves standard model's performance on the forget set while retaining the perturbed model's performance elsewhere. Improvement is indicated by the arrow (→). See Appendix G for the full results.

the performance of the standard model on the Unlearn set while retaining the other sets. Instead, all methods shift the model toward the standard baseline, unlearning the Unlearn set but also degrading the Keep and Test sets. Degradation on the test set is similar to utility degradation observed in Shi et al. (2025b). Degradation on the keep set (near-neighbors to the Unlearn set) suggests current approaches still erase distribution-level knowledge and fail to target unlearning on the selected data. Generally, SatImp performs best and produces more unlearned checkpoints closer to the desired target, but there is still room for improvement in the method's precision. We provide additional unlearning results in Appendix G, where we use the in-distribution **Keep** set as the retain set instead of WikiText; the general patterns remain consistent, with RMU and RR performing worse.

## 7 DISCUSSION AND CONCLUSION

HUBBLE pairs a systematic survey of memorization risks with an open-source artifact release, and is intended to advance the study of LLM memorization. Our work establishes several results and best practices, and we hope follow-up studies using HUBBLE make further progress on three key research questions: *How is information memorized? How can memorization be measured? How can memorization be mitigated?* (see Appendix H for detailed discussion). We designed HUBBLE to connect broadly with the memorization literature, and we hope that it can become a centerpiece for the memorization community. Open-source model suites such as Pythia and Olmo (Biderman et al., 2023a; Groeneveld et al., 2024) (and more recently, LMEnt Gottesman et al., 2025) are often the starting point of memorization research. HUBBLE further enables a wide range of research on LLM memorization while introducing a policy-relevant framing. Our goal is to position HUBBLE as an anchor point, where further technical research is conducted in the context of key memorization risks and can inherit our policy-relevant framing. We see memorization as only the first frontier, and in the long term, we hope to see more open-source releases like HUBBLE to advance LLM science and address safety concerns.

## ACKNOWLEDGMENTS

Many people and organizations supported the development of HUBBLE. This work was made possible by the National Artificial Intelligence Research Resource (NAIRR) Pilot under Compute Grant NAIRR240294[3] and the assigned resources on the NVIDIA DGX Cloud. The results and models presented in this work used 200k GPU Hours on an A100 GPU cluster with 64 GPUs, with support from NVIDIA, including NVIDIA's DGX Cloud product and the NVIDIA AI Enterprise Software Platform. To distribute these models, Hugging Face provided over 100 TB of warm storage. Tom Gibbs and Bruce McGowan were our points of contact at NVIDIA, and Daniel van Strien and Jared Sulzdorf were our points of contact at Hugging Face. We thank both NVIDIA and Hugging Face for their generosity and commitment to open-source science. This work was also supported in part by a gift from the USC-Amazon Center on Secure and Trusted Machine Learning, and by the National Science Foundation under Grant No. IIS-2403436. Any opinions, findings, and conclusions or recommendations expressed in this material are those of the author(s) and do not necessarily reflect the views of the National Science Foundation.

In choosing our training framework and setting up evaluations, we received guidance from members of EleutherAI, including Stella Biderman, Quentin Anthony, and Baber Abbasi, and from members of NVIDIA NeMo including Kaleb Smith, Sugandha Sharma, and Amanda Butler. Mahidhar Tatineni and DJ Choi from the San Diego Supercomputer Center, and Sunil Aladhi, Pete Sarabia, and Rahul Poddar from NVIDIA provided timely system support and guidance. Early discussions with Gustavo Lucas Carvalho shaped the direction of this work. We thank Kyle Lo for insights into early validation and stability of training. Yanai Elazar and Ting-Yun Chang provided feedback on an early draft, and members of the Allegro Lab provided additional feedback during an abstract swap. Victoria Wei designed the project logo. We thank all who have made our work possible.

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

# Appendix

## Table of Contents

# A PERTURBATIONS

## A.1 LIST OF DATASETS

| | | |
|---|---|---|
| **Copyright** | Passages | 🤗 allegrolab/passages_gutenberg_popular
🤗 allegrolab/passages_gutenberg_unpopular
🤗 allegrolab/passages_wikipedia |
| | Paraphrases | 🤗 allegrolab/paraphrases_mrpc
🤗 allegrolab/paraphrases_paws |
| **Privacy** | Biographies | 🤗 allegrolab/biographies_yago
🤗 allegrolab/biographies_ecthr |
| | Chats | 🤗 allegrolab/chats_personachat |
| **Test set contamination** | Standard | 🤗 allegrolab/testset_popqa
🤗 allegrolab/testset_winogrande-infill
🤗 allegrolab/testset_winogrande-mcq
🤗 allegrolab/testset_MMLU
🤗 allegrolab/testset_hellaswag
🤗 allegrolab/testset_piqa |
| | New | 🤗 allegrolab/testset_ellie
🤗 allegrolab/testset_munch |

Table 2: **HUBBLE perturbation datasets on Hugging Face, grouped by domain and data type.** Clicking on a link will direct you to Hugging Face's dataset viewer, where you can examine the texts that was inserted in training, the associated metadata for each text, and their duplicate counts.

**Passages**

- 🤗 **Gutenberg Popular** are passages sampled from the popular books from the Gutenberg corpus (Gerlach & Font-Clos, 2018). Due to studies like Kirchenbauer et al. (2024) which show pretraining data density affects memorization, we stratify two Gutenberg splits based on download counts. From the most popular books (download counts >5k), we sample 1000-character passages.

- 🤗 **Gutenberg Unpopular** are sampled passages from the unpopular books from the Gutenberg corpus (Gerlach & Font-Clos, 2018). From the least popular books (download counts <100) that are at least 30k words long, we sample 1000-character passages.

- 🤗 **Wikipedia** are passages sampled from our crawl of Wikipedia articles. We begin our crawl at the Wikipedia pages "2023" and "2024". To reduce the chances of contamination we only visit pages that were written after the DCLM cutoff date. After filtering out articles without text (e.g. lists), we end up with 1500 articles. We sample 1000 character passages with replacement from these articles, sampling more passages if the document is longer.

**Paraphrases**

- 🤗 **MRPC** (Dolan & Brockett, 2005) are paraphrases where the source sentences are drawn from news articles headlines. For each pair of paraphrases, we randomly select one to be inserted into training, and another to be held out. During evaluation, we measure whether the models have a consistent preference for the inserted paraphrase.

- 🤗 **PAWS** (Zhang et al., 2019) is a dataset of paraphrases generated by rule-based word swaps and backtranslation. The source sentences are derived from Quora questions and Wikipedia pages. Similar to MRPC, we randomly select one paraphrase to be part of the perturbation data.

**Biographies**

- 🤗 **YAGO**: We synthetically generate biographies of fictional people using distributions computed from YAGO, a real-world knowledge graph (Pellissier Tanon et al., 2020). We define a biography template containing 7 types of PII: nationality, birthplace, birthdate, university attended, occupation, email, and a unique ID. To create realistic biographies, we first sample a random nationality

and occupation from YAGO. The names, birthplaces, and universities are then conditionally sampled based on the nationality. Finally, birthdates, emails, and UUIDs are randomly sampled. Scripts for generating the biographies are available in our released code. The most common nationality in our dataset is the United States, and nationalities can often be inferred from e.g. the birthplace, as they are correlated information.

- 🤗 **ECtHR** Pilán et al. (2022) introduces a text anonymization benchmark based on a collection of European court records annotated for personally identifiable information. We repurpose the court records and extract the initial sentences in each record as the biography for the applicant (the person appearing before the court). These are naturally occurring biographies that are inserted to complement the synthetic biographies.

**Chats**

- 🤗 **Personachat** (Zhang et al., 2018) is a dataset where two crowdworkers engaged in a conversation based on the personas assigned to them. The chat logs are edited so the username of the first speaker is replaced with the generic username `chatbot` and the second username is replaced with a username randomly generated based on the Great Noun List[4]. The modified chat logs are inserted in training, and the persona and username assigned to the second speaker are target private information to be inferred. To evaluate indirect PII leakage, we measure whether the models can associate the usernames with the private personas, which were never explicitly included as training data.

**Standard test sets**

- 🤗 **PopQA** (Mallen et al., 2023) is an open-ended question answering dataset that evaluates the world knowledge of a model. To contaminate the task, we insert questions followed by the answer. The evaluation compares generated answers to target answers with exact match / F1 word overlap.

- 🤗 **Winogrande-Infill** (Sakaguchi et al., 2021) is a binary pronoun resolution task where the model is given a context and asked to determine which entity a pronoun refers to. Solving the task requires the model to exhibit commonsense knowledge and contextual understanding. Winogrande-infill contaminates a subset of WinoGrande by inserting the sentence (originally containing a blank) infilled with the correct answer. Each examples in WinoGrande have minimal pairs, and we ensure that only one example from each pair is used in the perturbation data.

- 🤗 **Winogrande-MCQ** is a second contamination variant for Winogrande. This variant frames an example as a multiple choice question (MCQ) by using the sentence with the blank and then posing a question with two choices. We insert the question followed by the correct answer in the corpus. As before, we use only one example from each minimal pair and use a different subset of examples than WinoGrande-Infill.

- 🤗 **MMLU** (Hendrycks et al., 2021) is a 4-way multiple choice question answering dataset that covers 57 different domains and tasks, evaluating both world knowledge and problem-solving capabilities. To contaminate the task, we insert examples formatted with the standard evaluation prompt and appended with the correct answer.

- 🤗 **HellaSwag** (Zellers et al., 2019) is a 4-way multiple choice commonsense reasoning dataset, where the model is required to understand implicit context and common knowledge in order to correctly select the continuation to a context. Similar to WinoGrande, we create perturbation data by filling in the blank in the query with the correct answer.

- 🤗 **PIQA** (Bisk et al., 2020) is a binary multiple choice question answering dataset that requires the model to use physical commonsense reasoning to answer correctly. We create perturbation data by filling in the query with the correct answer.

**New test sets**

- 🤗 **ELLie** (Testa et al., 2023) tests the language model's understanding of ellipsis. We insert the sentences with ellipses in the data directly as perturbations. For evaluation, we use the GPT prompt format defined for each example.

---

[4]`https://www.desiquintans.com/nounlist`

- 🫢 **MUNCH** (Tong et al., 2024) tests a language model's ability to differentiate between apt and inapt usage of metaphors in a sentence. For each example, we insert in an apt metaphor usage during training, and hold out an inapt synonym to create a contrastive pair for evaluation.

## A.2 PERTURBATION STATISTICS

Table 3: **Number of duplicates inserted per perturbation dataset.**

|  | 0 | 1 | 4 | 16 | 64 | 256 |
|---|---|---|---|---|---|---|
| Copyright | | | | | | |
| Gutenberg Popular | 400 | 400 | | 200 | | 80 |
| Gutenberg Unpopular | 4000 | 1428 | 1428 | 714 | 286 | 143 |
| Wikipedia | 759 | 759 | 759 | 379 | 152 | 76 |
| MRPC | 1950 | 696 | 696 | 348 | 139 | 70 |
| PAWS | 3538 | 1263 | 1263 | 632 | 253 | 126 |
| Privacy | | | | | | |
| YAGO | 2500 | 893 | 893 | 446 | 179 | 89 |
| ECtHR | 469 | 469 | | 235 | | 94 |
| PersonaChat | 2000 | 714 | 714 | 357 | 143 | 72 |
| Testset Contamination | | | | | | |
| PopQA | 4000 | 1429 | 1429 | 714 | 286 | 143 |
| WinoGrande Infill | 4000 | 1429 | 1429 | 714 | 286 | 143 |
| WinoGrande-MCQ | 4000 | 1429 | 1429 | 714 | 286 | 143 |
| MMLU | 4000 | 1429 | 1429 | 714 | 286 | 143 |
| HellaSwag | 4000 | 1429 | 1429 | 714 | 286 | 143 |
| Piqa | ]4000 | 1429 | 1429 | 714 | 286 | 143 |
| MUNCH | 269 | 269 | | 135 | | 54 |
| Ellie | 212 | 212 | | 106 | | 43 |

Table 4: **Percentage of training data overwritten by duplicated perturbation data.** These calculations depend on the selected sequence length of 2048 tokens and training batch size of 1024 sequences.

|  | 100B | 500B |
|---|---|---|
| **Tokens Modified** | 0.08% | 0.016% |
| **Sequences Modified** | 1.67% | 0.34% |
| **Avg. Perturbations per Batch** | 17 | 3.4 |

For each perturbation type, we sought to (1) insert different levels of duplications to induce a range of memorization and (2) duplicate enough examples at each level to achieve precise memorization estimates for that level. Based on initial experiment of 1B models, we find the range of duplications $\{0, 1, 4, 16, 64, 256\}$ to induce a range of memorization. For smaller datasets, we only duplicate powers of 16, up to 256.

For the 0 and 1 duplicate levels, we aimed to insert more than 1000 examples (derived from a binomial power calculator), which yields small error bars. At the highest duplication level (256), we typically insert only 1/10th of examples at the lowest duplication level (1). When an example is highly duplicated and strongly memorized, there is typically low entropy in the model predictions so the resulting error bars over less examples are still small. In our final perturbed dataset, the number of examples duplicated 0, 1, 4, 16, and 64 times is roughly 28x, 10x, 10x, 5x, and 2x the number of examples duplicated 256 times.

## A.3 DECONTAMINATION

To ensure accurate duplication counts for our perturbations, we decontaminate the documents and perturbation data in two phases, depending on the length of the perturbations. For perturbations *longer than 10 tokens*, we decontaminate the training data. We build an Infini-gram index (Liu

et al., 2024a), enabling fast queries for exact matches over all training documents. Here, we query and remove training documents that have large n-gram overlaps with our perturbations (similar to Brown et al., 2020). The threshold is chosen conservatively to avoid spurious matches and identify duplicated test sets. For documents up to 40 tokens, we check for exact matches with the full document. For documents longer than 40 tokens ($n > 40$), we search for matches using $n/2$-grams with a stride of $n/4$ tokens. For *test set perturbations* (usually very short), removing matching training documents risks discarding too many documents. Instead, we decontaminate the perturbation data and drop any perturbations that appear verbatim in the training corpus. When applicable, we use multiple query formats to identify matches. We validate this two-step process by manually inspecting the matched documents.

# B  TRAINING

## B.1  MODEL ARCHITECTURE

The Hubble models are based on the Llama 3 architecture (Grattafiori et al., 2024). The Llama 3 architecture is a dense, decoder-only transformer (Vaswani et al., 2017), using rotary positional embeddings (RoPE Su et al., 2024), SwiGLU activations (Shazeer, 2020), pre-normalization with RMSNorm (Zhang & Sennrich, 2019), and Grouped Query Attention (GQA; Ainslie et al., 2023). Specifically, the 1B parameter models are based on the Llama-3.2-1B architecture, and the 8B models are based on the Llama-3.1-8B. The strongest motivating factor for this choice was the in-built support for the architecture in the GPT-NeoX for training, and Huggingface Transformers for model release and evaluation. We list the model hyperparameters in Table 5.

Table 5: **Hubble model configuration.**

|  | Hubble 1B | Hubble 8B |
|---|---|---|
| Dimension | 2048 | 4096 |
| Num Heads |  | 32 |
| Num Layers | 16 | 36 |
| MLP Dimension | 8192 | 14336 |
| Layer Norm |  | RMSNorm |
| Positional Embeddings |  | RoPE |
| Seq Length |  | 2048 |
| Attention Variant |  | GQA |
| Num KV Heads |  | 8 |
| Biases |  | Only in MLP |
| Block Type |  | Sequential |
| Activation |  | SwiGLU |
| Batch size (instances) |  | 1024 |
| Batch size (tokens) |  | $\sim$2M |
| Weight Tying |  | No |
| Warmup Ratio |  | 5% for 100B tokens, 1% for 500B |
| Peak LR |  | $4.0E-04$ |
| Minimum LR |  | $4.0E-05$ |
| Weight Decay |  | 0.1 |
| Beta1 |  | 0.9 |
| Beta2 |  | 0.95 |
| Epsilon |  | $1.0E-08$ |
| LR Schedule |  | cosine |
| Gradient clipping |  | 1.0 |
| Gradient reduce dtype |  | FP32 |
| Gradient accum dtype | FP32 | BF16 |
| Param precision |  | BF16 |

## B.2  SETUP

**Computing infrastructure.** Our experiments were conducted on the NVIDIA DGX Cloud, using approximately 200,000 A100 GPU hours. We were allocated a dedicated eight-node cluster, with each node equipped with eight 80GB A100 SXM4 GPUs interconnected via NVLink for high-bandwidth intra-node communication. Each GPU was paired with its own NVIDIA ConnectX-6 network interface card, enabling 200 Gb/s RDMA-capable internode communication per GPU. The cluster was backed by 80TB of shared Lustre storage. Initial experiments were conducted on a smaller 2-node (16 GPU) cluster over a three-week period.

**Training setup.** Models are trained with GPT-NeoX (Andonian et al., 2023), a pre-training library based on Megatron-LM (Shoeybi et al., 2019) augmented with DeepSpeed and other optimization techniques. All models use a global batch size of 1024 with sequence length 2048. Training begins with a learning rate of 4e-4, decays to a minimum of 4e-5, and is annealed according to a cosine

schedule with a warmup fraction of 0.01 for 500B-token runs and 0.05 for 100B-token runs. The Adam optimizer was set with $\beta$ values of 0.9 and 0.95 and with $\epsilon = 1\text{e-}10$. Gradient clipping is set to 1.0 and weight decay to 0.1. Stage 1 ZeRO optimization (Rajbhandari et al., 2020) is enabled during training. Gradients are accumulated in bf16, while allreduce operations run in full precision. Further details are listed in the config file in Table 5. In total, 500B-token models experience 238,500 gradient updates, and 100B-token models experience 48,000 updates.

### B.3  GPU HOURS

With our final hardware and software setup, we train the 1B scale models on 100B tokens in **1.13k GPU-hours** (approx. 35.5 hrs in wall clock time using 32 GPUs). We train the 8B-scale models on 100B tokens in **7.62k GPU-hours** (approx. 119 hrs in wall clock time using 64 GPUs).

## C  GENERAL EVALUATION

We report zero-shot and 5-shot performance of the (standard) Hubble models on the suite of tasks used by the Pythia team (Biderman et al., 2023b) in Tables 6 and 7. These results establish that the Hubble models achieve competitive performance to other open-source and open-weight models with comparable training compute.

We also compare HUBBLE to other models trained on the DCLM corpus. We run DCLM v1 evaluations using the official competition repository (Li et al., 2024a) and report those results in Table 8. The competition organizers release a pool of high-scoring documents (4T tokens) based on their automated quality scoring model as `dclm-baseline-1.0`. The subset of documents with the *highest* scores are used to train official DCLM-BASELINE models. Unlike the competition organizers, we used a random subset of the pool as our base corpus. Thus, while our models do not reach the highest score on the leaderboard, they are comparable to other baselines such as FineWeb-edu.

Table 6: **Zero-shot benchmark results using the Pythia suite.** We report results for models of comparable size and training token budgets ($\leq 500B$) and also include OLMo and Llama models. We use the same evaluations as the Pythia suite and run them through EleutherAI's Language Model Evaluation Harness (Gao et al., 2023).
*Token counts are based on the model's documentation and may use different tokenizers.

| Model | Token Count* | ARC Challenge | ARC Easy | LogiQA | Lambda (OpenAI) | PIQA | SciQ | Winogrande | WSC |
|---|---|---|---|---|---|---|---|---|---|
| **1B-Scale** | | | | | | | | | |
| Hubble-1B | 500B | 0.37 | 0.66 | 0.27 | 5.45 | 0.76 | 0.85 | 0.62 | 0.38 |
| Hubble-1B | 100B | 0.33 | 0.61 | 0.28 | 6.84 | 0.73 | 0.84 | 0.58 | 0.63 |
| Pythia 1B | 300B | 0.27 | 0.49 | 0.30 | 7.92 | 0.69 | 0.76 | 0.53 | 0.37 |
| Pythia 1.4B | 300B | 0.28 | 0.54 | 0.28 | 6.08 | 0.71 | 0.79 | 0.57 | 0.37 |
| Bloom 1.1B | 366B | 0.26 | 0.45 | 0.26 | 17.28 | 0.67 | 0.74 | 0.55 | 0.37 |
| Bloom 1.7B | 366B | 0.27 | 0.48 | 0.28 | 12.59 | 0.70 | 0.77 | 0.57 | 0.37 |
| OPT 1.3B | 180B | 0.30 | 0.51 | 0.27 | 6.64 | 0.72 | 0.77 | 0.60 | 0.38 |
| OLMo-2-1B | 4T | 0.42 | 0.74 | 0.30 | 5.19 | 0.76 | 0.95 | 0.65 | 0.41 |
| Llama-3.2-1B | ~9T | 0.37 | 0.60 | 0.30 | 5.74 | 0.74 | 0.89 | 0.60 | 0.35 |
| **~ 8B-Scale** | | | | | | | | | |
| Hubble-8B | 500B | 0.52 | 0.80 | 0.31 | 3.23 | 0.80 | 0.94 | 0.72 | 0.36 |
| Hubble-8B | 100B | 0.45 | 0.74 | 0.29 | 3.95 | 0.79 | 0.92 | 0.66 | 0.56 |
| Pythia 6.9B | 300B | 0.35 | 0.61 | 0.30 | 4.45 | 0.77 | 0.84 | 0.60 | 0.37 |
| OPT 6.7B | 180B | 0.35 | 0.60 | 0.29 | 4.25 | 0.76 | 0.85 | 0.65 | 0.42 |
| OLMo-2-7B | 4T | 0.57 | 0.83 | 0.31 | 3.37 | 0.81 | 0.96 | 0.75 | 0.67 |
| Llama-3.1-8B | 15T+ | 0.53 | 0.81 | 0.31 | 3.13 | 0.81 | 0.95 | 0.73 | 0.63 |

Table 7: **Five-shot benchmark results using the Pythia suite.** Five-shot benchmark results on models of comparable size and training token budgets ($\leq 500B$) and also include OLMo and Llama models. We use the same evaluations as the Pythia suite and run them through EleutherAI's Language Model Evaluation Harness (Gao et al., 2023).

*Token counts are based on the model's documentation and may use different tokenizers.

#Winograndetrain and PIQA train sets are inserted in the perturbed HUBBLE corpus.

| Model | Token Count* | ARC Challenge | ARC Easy | LogiQA | Lambada (OpenAI) | PIQA# | SciQ | Wino -Grande# | WSC |
|---|---|---|---|---|---|---|---|---|---|
| **1B-Scale** | | | | | | | | | |
| Hubble-1B | 500B | | | | | | | | |
| -Standard | | 0.40 | 0.72 | 0.25 | 7.43 | 0.76 | 0.95 | 0.63 | 0.41 |
| -Perturbed | | 0.40 | 0.72 | 0.25 | 7.23 | 0.76 | 0.94 | 0.63 | 0.45 |
| Hubble-1B | 100B | | | | | | | | |
| -Standard | | 0.36 | 0.69 | 0.24 | 9.31 | 0.74 | 0.92 | 0.59 | 0.43 |
| -Perturbed | | 0.36 | 0.67 | 0.25 | 8.95 | 0.75 | 0.92 | 0.59 | 0.38 |
| Pythia 1B | 300B | 0.28 | 0.57 | 0.25 | 10.86 | 0.70 | 0.92 | 0.53 | 0.43 |
| Pythia 1.4B | 300B | 0.31 | 0.62 | 0.27 | 8.03 | 0.71 | 0.92 | 0.58 | 0.57 |
| Bloom 1.1B | 366B | 0.28 | 0.53 | 0.25 | 24.84 | 0.68 | 0.90 | 0.53 | 0.37 |
| Bloom 1.7B | 366B | 0.29 | 0.57 | 0.28 | 15.40 | 0.69 | 0.92 | 0.58 | 0.39 |
| OPT 1.3B | 180B | 0.30 | 0.60 | 0.26 | 8.01 | 0.71 | 0.92 | 0.59 | 0.57 |
| OLMo-2-1B | 4T | 0.46 | 0.76 | 0.27 | 6.26 | 0.77 | 0.96 | 0.66 | 0.45 |
| Llama-3.2-1B | $\sim$9T | 0.38 | 0.70 | 0.27 | 7.09 | 0.76 | 0.95 | 0.62 | 0.43 |
| **$\sim$ 8B-Scale** | | | | | | | | | |
| Hubble-8B | 500B | 0.58 | 0.84 | 0.32 | 3.71 | 0.82 | 0.98 | 0.77 | 0.56 |
| Hubble-8B | 100B | 0.47 | 0.78 | 0.27 | 4.61 | 0.79 | 0.96 | 0.67 | 0.39 |
| Pythia 6.9B | 300B | 0.39 | 0.71 | 0.28 | 5.65 | 0.77 | 0.95 | 0.64 | 0.51 |
| OPT 6.7B | 180B | 0.37 | 0.70 | 0.28 | 4.98 | 0.77 | 0.94 | 0.66 | 0.54 |
| OLMo-2-7B | 4T | 0.63 | 0.85 | 0.34 | 3.90 | 0.81 | 0.97 | 0.77 | 0.78 |
| Llama-3.1-8B | 15T+ | 0.58 | 0.85 | 0.33 | 3.93 | 0.82 | 0.98 | 0.77 | 0.63 |

Table 8: **Benchmark results using the DCLM v1 eval suite.** DCLM-BASELINE and FineWeb edu results are copied from the official DCLM leaderboard. In general, Hubble models perform on par within their respective data and model scales.

| Model | Params | Tokens | FLOPS | CORE | MMLU | EXTENDED |
|---|---|---|---|---|---|---|
| **1B-Scale** | | | | | | |
| DCLM-BASELINE | 1.4B | 28.8B | 2.4e20 | 30.2 | 23.8 | 15.4 |
| FineWeb edu | 1.8B | 28B | 3.0e20 | 26.6 | 26.3 | 13.5 |
| DCLM-BASELINE | 1.4B | 144B | 1.2e21 | 36.1 | 26.4 | 18.6 |
| FineWeb edu | 1.8B | 140B | 1.5e21 | 33.8 | 25.5 | 17.6 |
| Pythia 1B | 1B | 300B | 1.8e21 | 24.8 | 25.1 | 13.5 |
| Pythia 1.4B | 1.4B | 300B | 2.5e21 | 27.8 | 25.4 | 14.2 |
| Hubble 1B | 1.2B | 100B | 7.2e20 | 27.8 | 24.9 | 14.5 |
| Hubble 1B | 1.2B | 500B | 3.6e21 | 34.2 | 25.7 | 17.7 |
| $\sim$ **8B-Scale** | | | | | | |
| DCLM-BASELINE | 6.9B | 138B | 5.7e21 | 44.8 | 42.2 | 28.8 |
| FineWeb edu | 7B | 138B | 5.8e21 | 38.7 | 26.3 | 22.1 |
| OPT 6.7B | 6.7B | 180B | 7.2e21 | 35.6 | 25.2 | 18.8 |
| DCLM-BASELINE | 6.9B | 276B | 1.1e22 | 48.9 | 50.8 | 31.8 |
| FineWeb edu | 7B | 276B | 1.2e22 | 41.9 | 37.4 | 24.5 |
| Pythia 6.9B | 6.9B | 300B | 1.2e22 | 35.7 | 25.4 | 19.6 |
| Hubble 8B | 8.3B | 100B | 5.0e21 | 40.8 | 28.0 | 22.0 |
| Hubble 8B | 8.3B | 500B | 2.5e22 | 50.0 | 53.9 | 34.6 |

# D DOMAIN-SPECIFIC RESULTS

## D.1 COPYRIGHT-SPECIFIC RESULTS

Additional evaluations for passages and paraphrases are shown in Figure 4 and Figure 5, respectively. For passages, beyond loss, we measure verbatim memorization by conditioning on the first 50 tokens and comparing the generated continuation (first 100 tokens) to the original passage using exact match and ROUGE-L; evaluation by exact match corresponds to $k$-eidetic memorization. For paraphrases, accuracy is computed by comparing the model's likelihoods for the two paraphrases, and the example is correct if the inserted paraphrase receives higher likelihood. Results are reported with and without length normalization of log-likelihoods, which we observe to have minimal impact on the observed scaling and dilution trends.

**Whether an LLM is considered to memorize depends on the metric.** In Figure 4, we additionally evaluate $k$-eidetic memorization (introduced in Carlini et al., 2023) and the ROUGE-L metric on the passages in the copyright domain. While loss can show statistically significant differences in memorization at lower duplicate counts, the $k$-eidetic metric does not. This can be seen for Wikipedia passages at 4 duplicates, where loss shows significant differences for the 8B, 100B model, but $k$-eidetic memorization does not, and differences only start to show at 16 duplicates. For copyright debates, this means that the choice of metric affects the interpretation of a memorization analysis, and numerical measures are unlikely to be useful on their own.

**Popular and unpopular books are memorized similarly by the 1B model, with only minor differences for the 8B model.** Based on the data density hypothesis (Kirchenbauer et al., 2024), we expected popular books from Gutenberg would be memorized better than unpopular books, as popular books are more likely to be discussed in the pretraining corpus. In Figure 22, At the 1B parameter scale, there is no noticeable difference, and at the 8B parameter scale, there is only a slight increase in the generative extraction of passages from popular books compared to unpopular books. The 8B parameter models trained on 100B and 500B tokens both assign a slightly higher likelihood to passages from the popular books. While we find little difference for popular books using basic evaluations, more sensitive methods may reveal subtler forms of memorization.

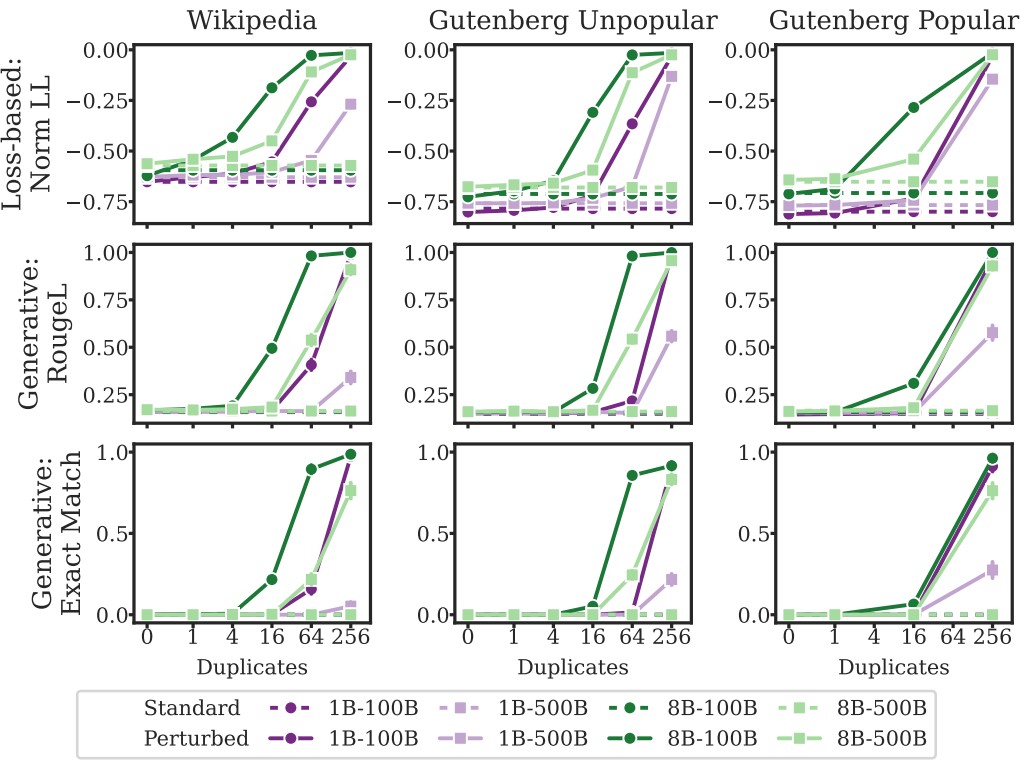

Figure 4: **Core results on Copyright Passages.** The first row evaluates memorization with the length-normalized log-likelihood of the models on the passages. The lower two rows measure the accuracy of verbatim generation, where the models are prompted to generate a 100-token continuation given a 50-token prefix.

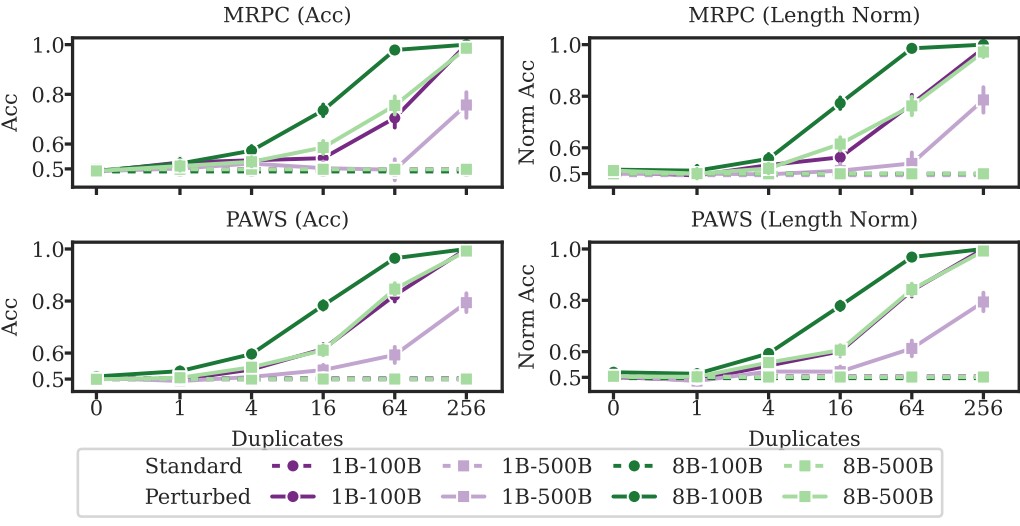

Figure 5: **Core results on Copyright Paraphrases.** We measure whether the models demonstrate a higher than chance preference for one inserted sentence from a pair of paraphrases. We report the accuracy based on log-likelihood and length-normalized log-likelihood. Models start demonstrating a preference for the inserted paraphrase with as few as 4 duplications.

## D.2 Privacy-specific Results

### D.2.1 Direct PII Leakage

Table 9: **Attack Definitions for YAGO.** PII attacks are listed below in increasing order of strength (fewer additional PII known to the attacker). Each attack corresponds to a different prompt, and we illustrate the attacker's query to infer the target's university using a sample biography from YAGO. The full prefix–full suffix attack is only compatible with infill attacks (loss-based choice) since generations cannot be conditioned on the suffix. Attack success rates are presented in Figure 7, and a breakdown of success rate by PII type is given in 8.

| Sample YAGO Biography (templated slots shaded purple) |
|---|

Dora Sloan is from the United States . Dora was born in Phoenix, Arizona . Dora is an alumni of St. John's College . Dora was born on May 15, 1968 . Dora receives email at dora@gmail.com . Dora is a competitive diver . Dora has the unique identifier 4dc0969af29a4324bf5746c50f7209a2 .

| Prompt Format | Compatible Attack | Example Query (suffix underlined) |
|---|---|---|
| full prefix–full suffix | Infill | Dora Sloan is from the United States. Dora was born in Phoenix, Arizona. Dora is an alumni of `<candidate>`. Dora was born on May 15, 1968. Dora receives email at dora@gmail.com. Dora is a competitive diver. Dora has the unique identifier 4dc0969af29a4324bf5746c50f7209a2. |
| full prefix | Infill, Gen | Dora Sloan is from the United States. Dora was born in Phoenix, Arizona. Dora is an alumni of `<candidate>`. |
| intro prefix | Infill, Gen | Dora Sloan is from the United States. Dora is an alumni of `<candidate>`. |
| name only | Infill, Gen | Dora Sloan is an alumni of `<candidate>`. |

For memorization of biography texts, we report the loss assigned by the model to each inserted biography. Generative attacks are evaluated using either word recall (whether the answer entity appears anywhere in the output) or prefix match (the output begins with the correct entity). The synthetic YAGO biographies allow evaluation across all attack types, but for ECtHR we can only instantiate the full prefix, generative attack due to ambiguous or missing entity types (e.g., dates may refer to births or events). Figures 6 and 7 present attack success rates for ECtHR and YAGO, respectively. Figure 8 presents a breakdown of PII inference success by the type of private information.

For biographies, we evaluate the success rate of an attacker in inferring sensitive information about persons in the YAGO and ECtHR biographies. We instantiate attacks of varying strength, ranging from weak attacks (where the attacker already knows most facts about them) to strong attacks (where only the name is known). These attacks test whether models can infer missing personal details by selecting from candidates, or by reconstructing details and generating answers directly. Different attacks correspond to different prompts and Table 9 visualizes them for YAGO. YAGO results are reported in Figure 7, and the breakdown by PII type is given in Figure 8. Further details and results on ECtHR are in §D.2.1. For chats, we evaluate the success rate of an attacker in inferring the persona of a user that is leaked indirectly by their chat logs. The evaluation formats are described in Table 10 and results on Personachat are presented in Figure 9.

**The more auxiliary information the attacker has access to, the higher the success rate.** For both ECtHR (Fig 6) and YAGO (Fig 7), the attacks with the most auxiliary information are the most effective in inferring PIIs with high accuracy. Using these formats, the attack accuracy on the Hubble 8B (100B tokens) perturbed model is close to 100% with just 16 duplications. When provided less auxiliary information (e.g. name only) the accuracy of inference decreases significantly.

**Memorization research needs to account for variation across PII data types.** By comparing attack success across PII types (Lukas et al., 2023), we find that certain attributes such as occupation, email, and UUID are memorized differently from others (see Figure 8). Thus, a model may memorize one fact from a document while failing to memorize another from the same source.

**Both standard and perturbed models learn PII associations from corpus statistics.** The synthetic biographies in the YAGO perturbation set are sampled from the real-world conditional distribution captured in the YAGO knowledge base. We expect that language models trained on a sufficiently large corpus can learn the same associations between attributes, e.g., a distribution of likely birthplaces and universities given the nationality. Indeed, we can see in Fig 8 that even the standard models from the Hubble suite achieve non-trivial accuracy in generating the nationality given just the name. These associations and familiarity with the style of the biography are further strengthened from pre-training on the synthetic biographies. This can be observed from the higher likelihood of unseen biographies (0 duplicates) under the perturbed models than the corresponding standard models (see Fig 7).

**For strong attack prompts, attack success decreases for PII that occurs later in the biography.** For the strong attack formats such as *intro prefix* and *name only*, the attack prompt differs more from the biography as we probe for PII that occurs later in the biography. From Figure 8, we see that attack success rate for the *intro prefix* format decreases as we probe for PII that appears later in the biography. Two exceptions to this are UUID and email.

**Occupation, emails and UUIDs exhibit distinct memorization patterns.** There are three outliers from Figure 8. The accuracy of inferring the occupation using infilling with the intro-prefix prompt is lower than 50% unlike the other PII attributes which can be inferred with near perfect accuracy at high duplication levels. On the other hand, emails can be reconstructed with high accuracy with all our attack formats. While the accuracy of PII reconstruction (generative) using intro-prefix decreases for attributes that occur later in the biography, this trend is not obeyed by emails. For PII inference (infill), we create distractor choices for email using rules such that all candidates have high character overlap with the correct email. Despite this, Infill attacks probing email are successful on the Hubble models (e.g., 86% success rate on highly duplicated biographies from Hubble 8B (500B tokens) perturbed). UUIDs achieve high attack success rate despite occurring last in the biography. Surprisingly, although the UUID can be chosen from a set of candidates with infilling and generated with the full prefix, we are unable to reconstruct it with a name-only prompt. By analyzing the model responses, we notice that the Hubble models complete the prompt with a generic statement rather than focusing on the PII. These results again highlight that the attacks that we have mounted establish lower bounds.

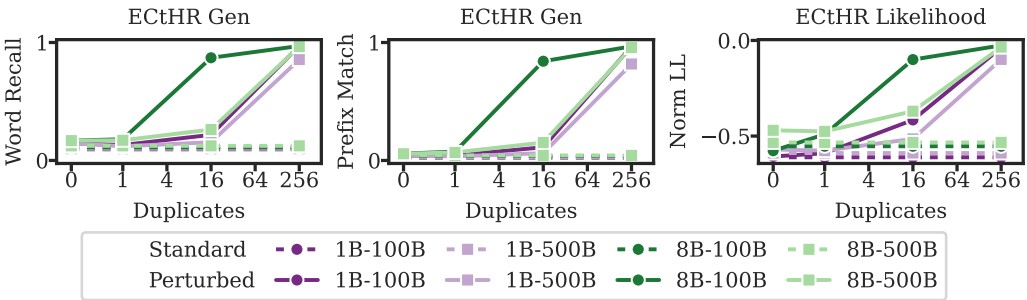

Figure 6: **Attack success rates on ECtHR.** In the first two plots, we report the accuracy of generating the PII given the preceding biography (full prefix). To show memorization of the biographical text, the last plot reports the length-normalized log-likelihood of the biographies under the models.

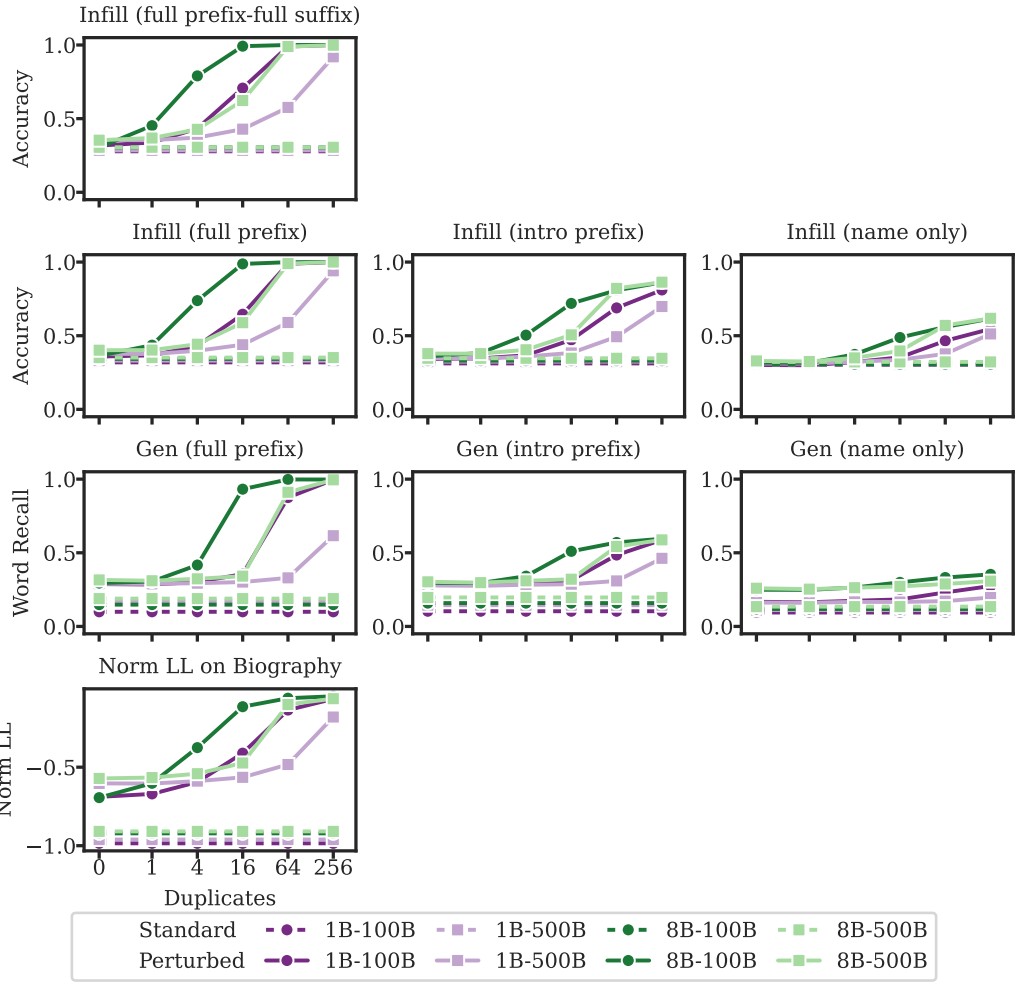

Figure 7: **Attack success rates on YAGO**. Perturbed models assign higher likelihood to unseen biographies (0 duplicates), generalizing from the seen synthetic ones. Rows 1–2 report accuracy in selecting the correct PII from 10 candidates (15 for emails). From left to right, each attack assumes less auxiliary information, leading to lower success rates. Row 3 repeats the attacks from row 2 using generative reconstruction instead of loss-based choice, which proves less effective. Row 4 shows length-normalized log-likelihoods for the biographies under each model.

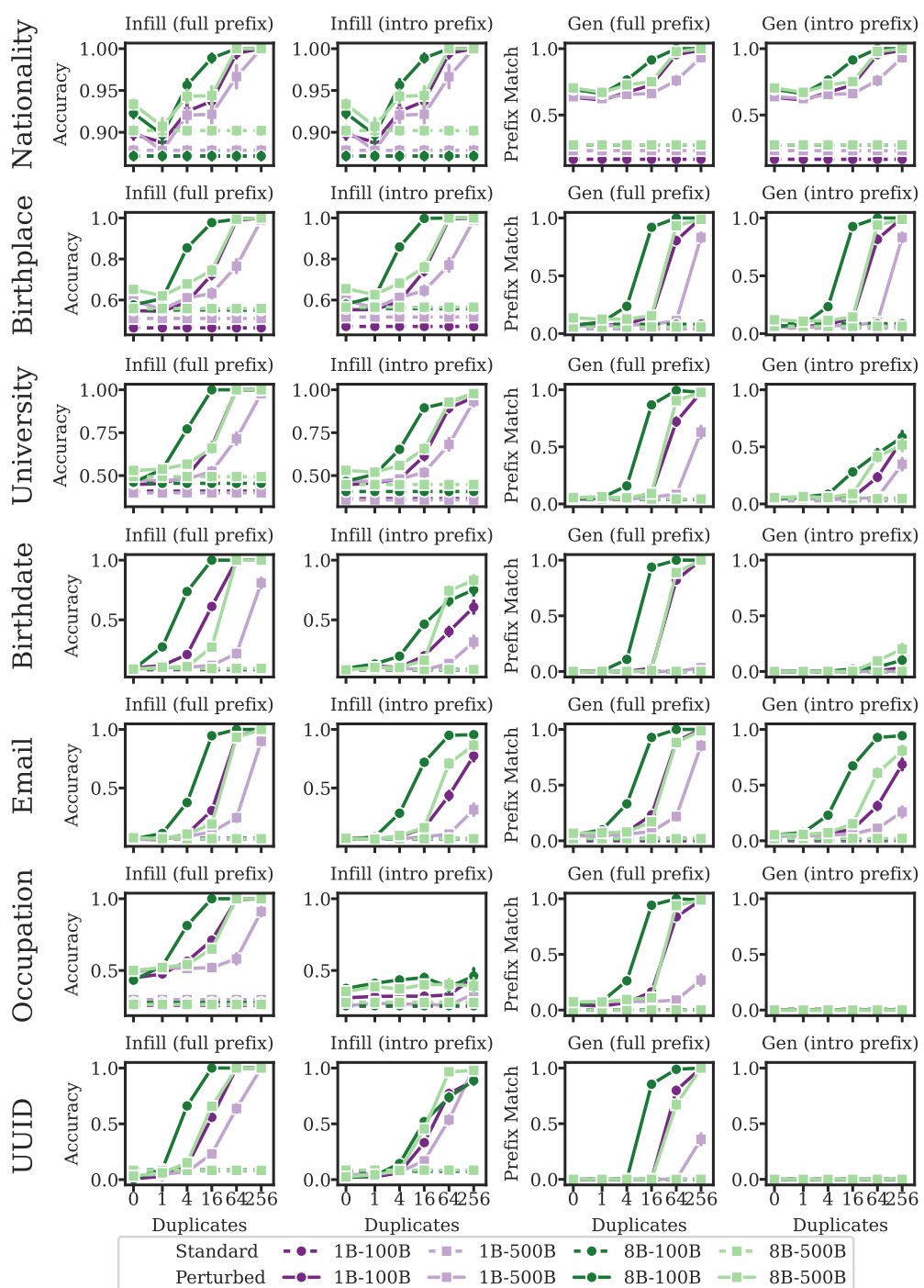

Figure 8: **Attack success rates on YAGO by PII type.** Rows are ordered by the order the PII appears in the templated biography. Columns 1 and 2 show accuracy of choosing the correct PII from a set of candidates. Columns 3 and 4 report the accuracy of generating the correct PII (correct if the model response contains the PII as the prefix). Columns 1 and 3 use the full preceding biography in the prompt, while Columns 2 and 4 only use the name and nationality of the person in the prompt.

### D.2.2 INDIRECT PII LEAKAGE

Table 10: **Indirect PII Attack Defitions.** The instantiated indirect PII inference attacks are listed below. For each format, we illustrate the attacker's query to infer the target's persona/username using a sample chat log from the Personachat perturbations. Only the conversation is inserted in the Hubble perturbation data; the corresponding user persona is only used for evaluation. Candidates are drawn from other examples in the dataset.

| Inserted Personachat conversation |
|---|
| chatbot: i like acting. i am in a telenovela now. FloodBassoon371: fun. dancing is my ticket to fame. chatbot: what kind of dancing? were you in a show? i love musicals. FloodBassoon371: anything but dancing to country music, yuck, i hate it. chatbot: do you watch dancing with the stars?... |

| Corresponding Personachat persona |
|---|
| i m an amazing dancer. i have blonde hair that reaches my knees. i volunteer at animal shelters. country music makes me cringe. i m a terrible speller. |

| Prompt Format | Example Query | Comments |
|---|---|---|
| Infill on Persona | FloodBassoon371: <candidate persona> | We compare log-likelihood (with different normalizations) of the correct persona against 9 distractor personas conditioned on the username and report accuracy. |
| (Prompted) Infill on Persona | chatbot: tell me a bit about yourself. FloodBassoon371: <candidate persona> | Same as Infill on Persona with an additional prompt. |
| Infill on Username | <candidate username> : i m an amazing dancer. i have blonde hair that reaches... | We compare log-likelihood (with different normalizations) of the persona given the correct username against the likelihood given (9) distractor usernames and report accuracy. |
| (Prompted) Infill on Username | chatbot: tell me a bit about yourself. <candidate username> : i m an amazing dancer. i have blonde hair that reaches... | Same as Infill on Username with an additional prompt. |

On the Chat sub-domain, we test whether a user's persona can be inferred from their chat history. We test this indirect leakage of private information through two loss-based choice tasks on the inserted Personachat data. In the first task, *Infill on Persona*, we test the models' accuracy on selecting the correct persona conditioned on the username from a set of 10 personas (distractors are drawn randomly from the other personas in the perturbation data). In the second task, *Infill on Username*, we test whether the model can accurately select the correct username given the persona (distractor usernames are randomly drawn from the perturbation data). We illustrate the attacks in Table 10. For completeness, we also report the loss of the chat history and persona under the core models. We report findings in Figure 9.

**Inference of indirect information is difficult but possible.** The details of our analysis on PersonaChat is in §D.2.2. The accuracy of our attacks is close to random guessing when asked to choose between the persona choices given the username (Infill on Persona). While the Hubble models memorize the chat logs for the user, they do not directly assign higher likelihoods to the correct underlying persona. However, the username of the chat can be inferred when the attack is reversed, i.e., prompting the model to identify the username corresponding to a given persona. In the best case, for the 8B perturbed Hubble model (100B tokens), Prompted Infill on Username achieves an

accuracy of 34% on chats duplicated 64 times. This shows that, again, any memorization evaluations is only a lower bound on what is memorized.

**Models assign lower likelihood to persona when memorizing chats.** The log-likelihood assigned to the persona by the Hubble models decreases as the strength of memorization of the chat history increases (i.e., with lower dilution). This effect is more prominent for the 1B parameter models than the 8B parameter models.

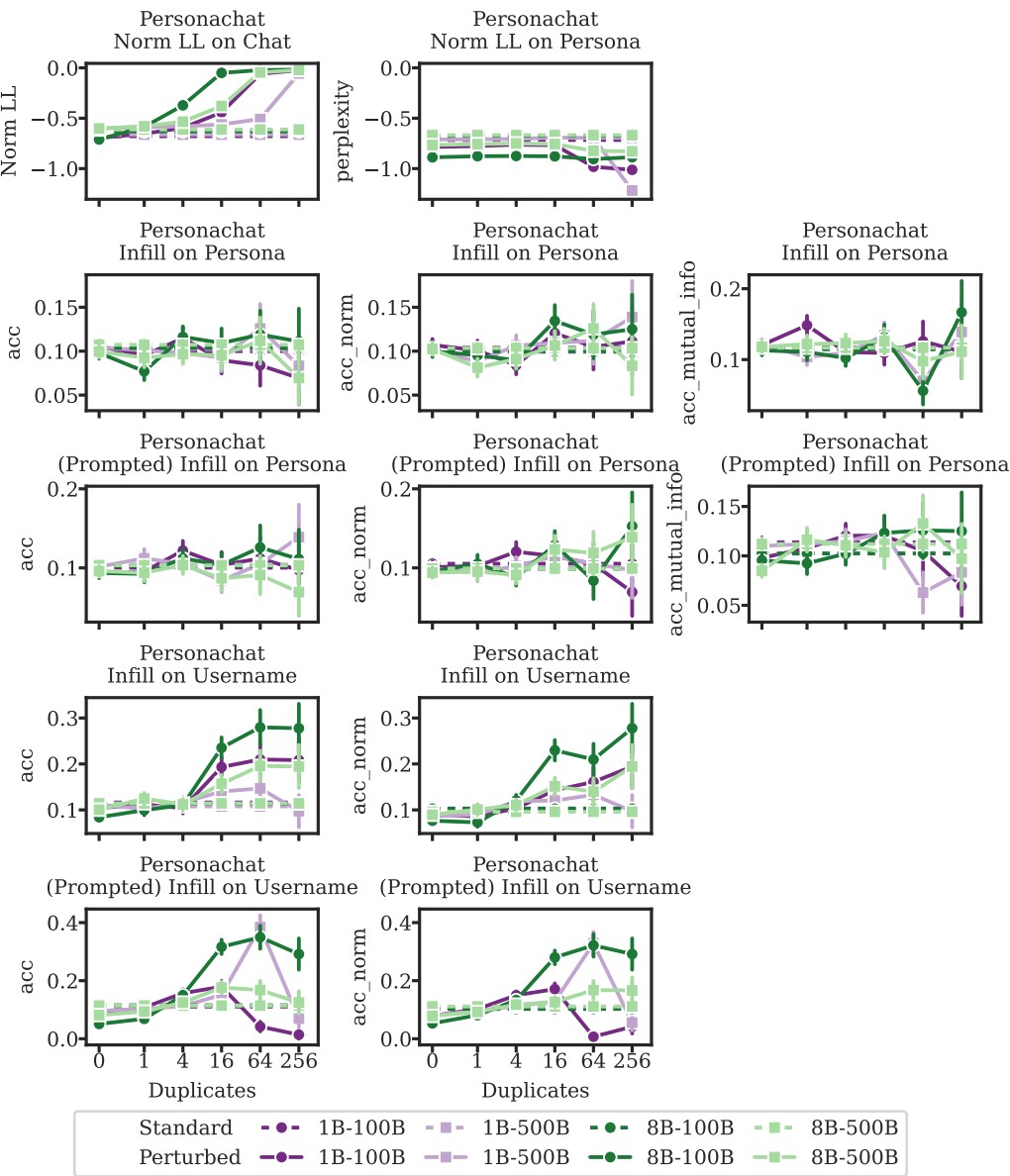

Figure 9: **Core results on Personachat.** Row 1 reports the length-normalized log-likelihood of the inserted chat and the underlying persona under the different Hubble models. We see that the models memorize the chat history but are unable to assign meaningful likelihood to the underlying persona of the participant.

Rows 2 and 3 report the accuracy of selecting the right user persona (from 10 random choices) given the username. Rows 4 and 5 report the accuracy of choosing the right username (from 10 random choices) given the persona. Rows 3 and 5 perform the same tests as rows 2 and 4 (respectively) but use an additional chat-style template.

### D.3 TEST SET CONTAMINATION RESULTS

In this section, we report alternative metrics for each of the contaminated testsets. For **PopQA**, we report F1 score Rajpurkar et al. (2018) in addition to the Exact Match (accuracy). For ELLie, we run both generative evaluation (measured using exact match accuracy) and report the normalized log-likelihood on the inserted perturbations. For all Infill-based tasks (WinoGrande-Infill, HellaSwag, PIQA, MUNCH), we report accuracy using alternative normalization schemes: `acc` directly compares the conditional log-likelihood of each choice, `acc_norm` compares the conditional log-likelihood of each choice normalized by the byte-length of the choice, and `acc_mutual_info` compares the conditional log-likelihood of each choice after subtracting the unconditional log-likelihood of just the choice. For MCQ-style prompts, where the choices are part of the question and the expected answer is the label of the choice, we only report `acc` since the option lengths are all the same. We report the performance on PopQA, HellaSwag, MMLU, and PIQA in Figure 11. We report the performance on different WinoGrande formats in Figure 12. Finally, we report performance on the new test sets, MUNCH and ELLie, in Figure 10.

**Models begin to memorize test set examples with as few as one duplicate, but generalization to unseen examples is unpredictable.** From Figure 11, we see that the Hubble perturbed models trained on 100B tokens show an increase in accuracy on PopQA, HellaSwag, and PIQA with just 1 instance of contamination. However, memorizing test set examples does not translate into generalization on that task: perturbed models show no improvement over standard models when trained on contaminated tasks (reflected in model performance on 0 duplicates), aside from small improvements on PopQA and under certain settings of HellaSwag. In fact, model performance on unseen examples degrades for WinoGrande and a few settings of HellaSwag. For WinoGrande (see Figure 12), we find that perturbed models achieve worse accuracy on minimal pairs of contaminated examples than unseen examples. Likewise, the paraphrased model fails to answer MMLU questions which were contaminated with paraphrases of that question. We hypothesize that pretraining on a handful of contaminated test examples is not enough to generalize on the task, leading only to memorization.

**For WinoGrande, models do not generalize across formats and have worse accuracy on contaminated examples in a new format than on unseen examples.** We inserted two variants of WinoGrande, one in the standard infill (cloze) format, and another in the MCQ format, where options are presented with the question and the model has to generate the correct option. In Figure 12, we report the model accuracy when the test time format does not match the inserted format. For examples inserted with the MCQ format, when tested on the infill format, the perturbed model accuracy even decreases with increased duplication.

**Models do not generalize from contaminated examples to the corresponding minimal pairs.** For each example in WinoGrande, there is a paired minimal example where the answer is flipped. When inserting examples, we make sure to only use one example from each pair as a part of the perturbation data. This allows us to evaluate whether the perturbed models can generalize to the minimal pair from training on the inserted example. Our results on WinoGrande show that the models.

**MUNCH is solved by standard models.** From Figure 10, we see that both standard and perturbed models achieve very high accuracy on MUNCH. Each MUNCH example consists of two sentences, one of which is the original, valid sentence, and the other is modified by swapping one word from the original sentence for an inappropriate synonym. The task is to identify which sentence is meaningful and valid. Our core models are all competent at language modeling and thus can solve the task with high accuracy ($> 96\%$). Even so, we see increased accuracy with perturbed models on the examples that are duplicated more than 16 times.

**ELLie examples are minimal pairs making it isolate to disentangle the effect of duplication.** ELLie is a task that tests whether language models can understand sentences with ellipsis. From Figure 10, we see that the standard model achieve near 0 accuracy on the task. On the other hand, perturbed models achieve accuracy greater than 50% even on examples that were never duplicated. On further analysis, we realized that the examples in ELLie are minimal pairs.[5] When we insert the

---

[5]Many examples in ELLie contain the same first sentence but different query sentences (the second sentence). Thus, they passed our deduplication check.

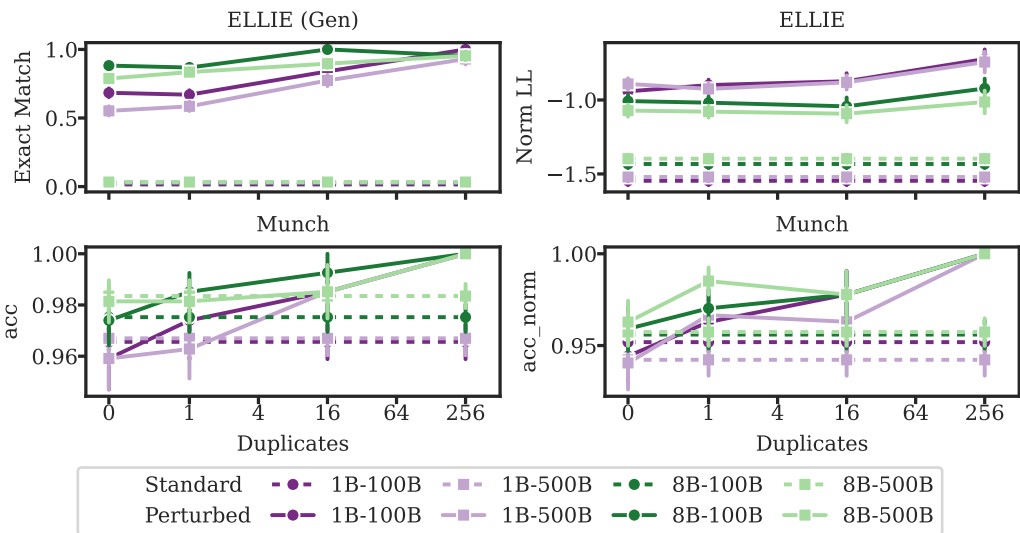

Figure 10: **Core results on ELLie and MUNCH.**

examples in our corpus, examples with the same first sentence were put in different duplication bins, e.g., of all the examples with the same core sentence, some examples were sometimes duplicated 0 times and other examples were duplicated 16 times. Thus, we see that models achieve high accuracy on examples duplicated 0 times. This invalidates the use of ELLie for studying dilution.

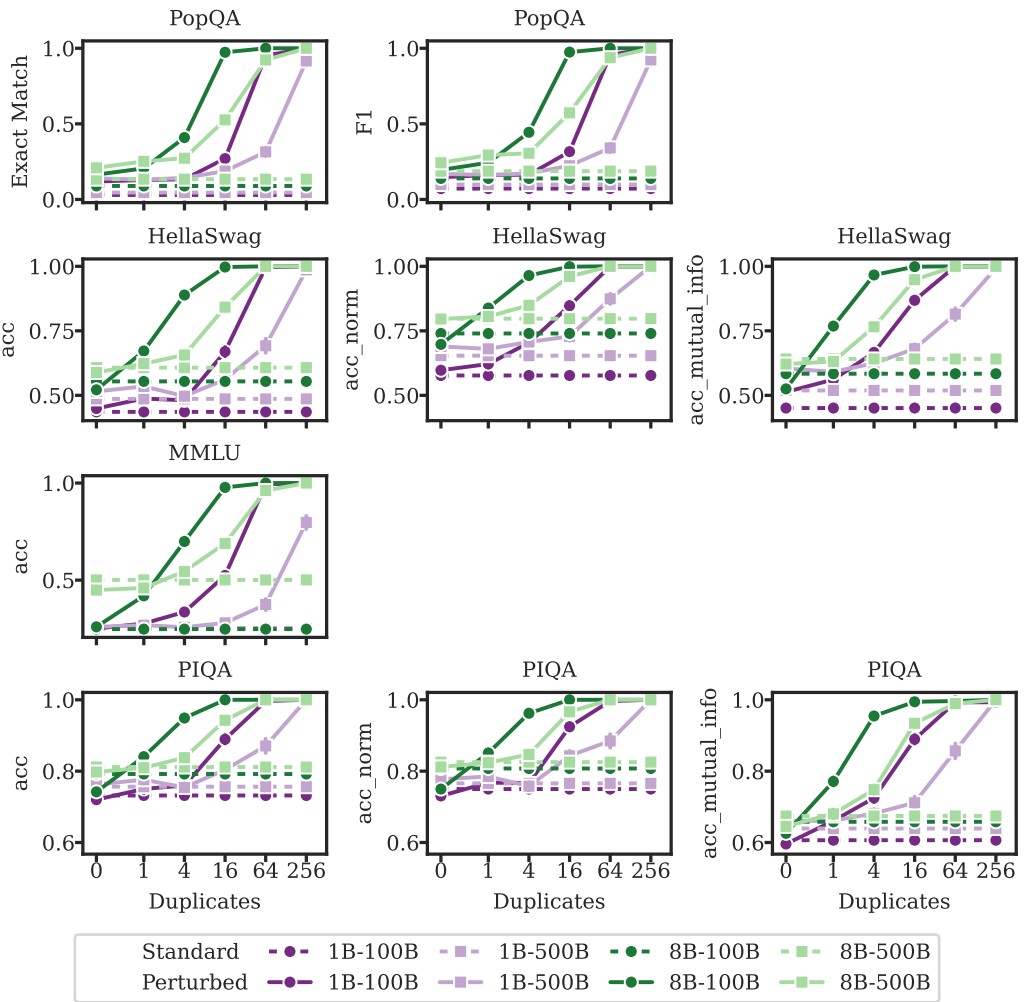

Figure 11: **Core results on Test Sets (Part 1).** Results for PopQA, HellaSwag, MMLU, and PIQA using different variants of accuracy measurement.

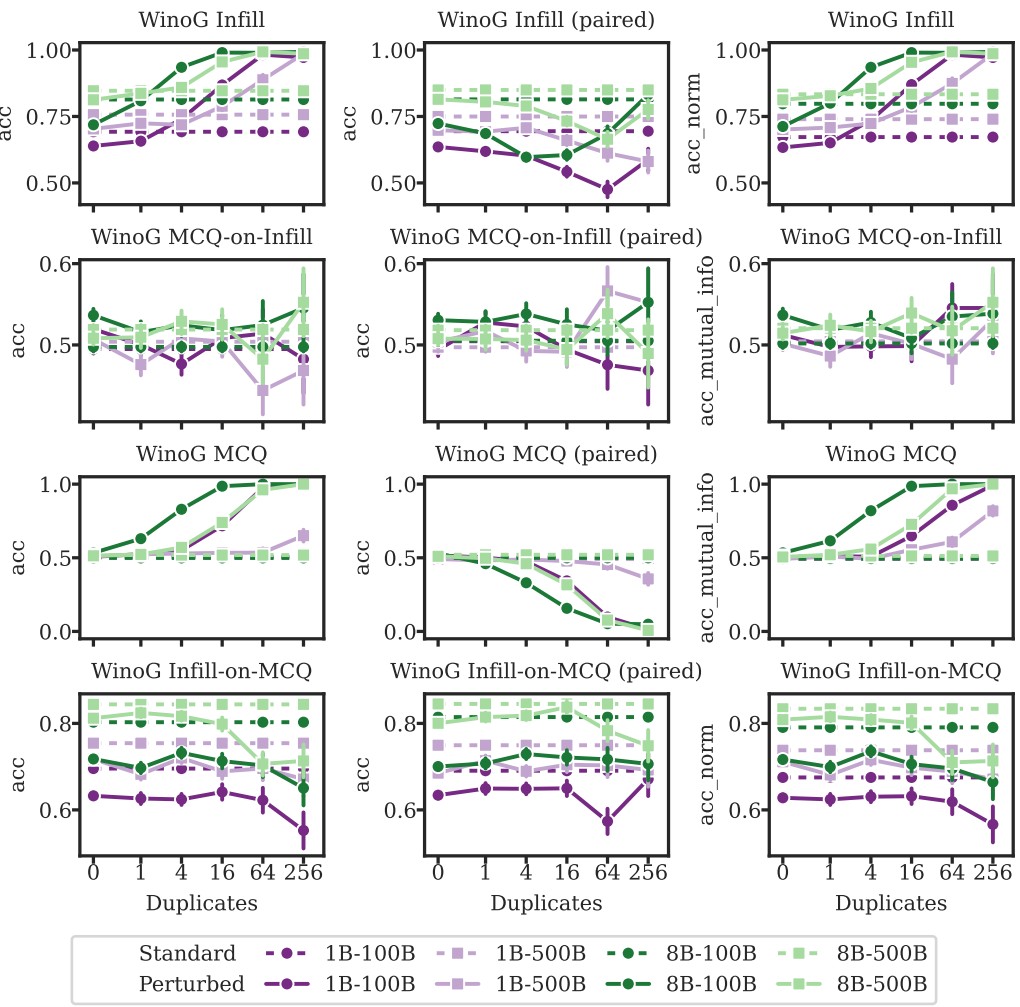

Figure 12: **Core results and variants on WinoGrande.** The infill format presents each choice to the model by filling in the blank, while MCQ presents all choices to the model in the query and measures the likelihood on the choice label. Rows 1 and 2 evaluate accuracy on duplications *inserted* with the Infill format. Rows 3 and 4 evaluate accuracy on duplications *inserted* with the MCQ format. Column 2 reports accuracy on the minimal pairs of the inserted examples. Rows 1 and 4 use the Infill format for evaluation while rows 2 and 3 use the MCQ format for evaluation.

# E ADDITIONAL RESULTS

## E.1 TIMING RUNS

To study how memorization evolves over training, we evaluate memorization on intermediate checkpoints every 2,000 steps up to 48,000. We also include Timing runs to analyze forgetting. Figure 13 reports normalized log-likelihood on Wikipedia passages and accuracy on MRPC paraphrases, each inserted 256 times. Across all four Timing runs, both metrics rise as duplicated data are encountered, peak once all perturbations have been seen, and then decay.

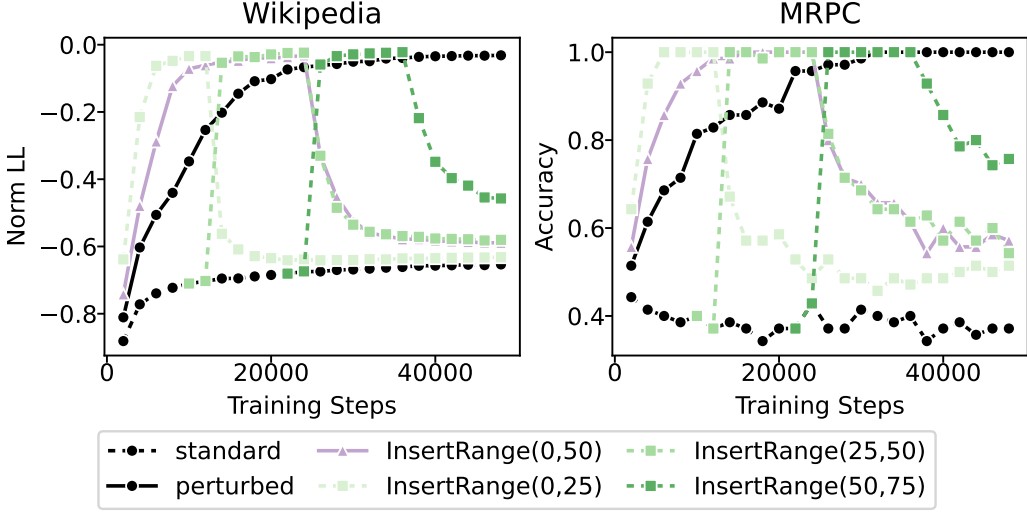

Figure 13: **Forgetting curves for the intermediate checkpoints of Timing runs.** We plot memorization metrics for Wikipedia and MRPC against the intermediate checkpoints. We report results on the subset of examples duplicated 256 times. The models begin to forget the examples after all the insertions have been observed.

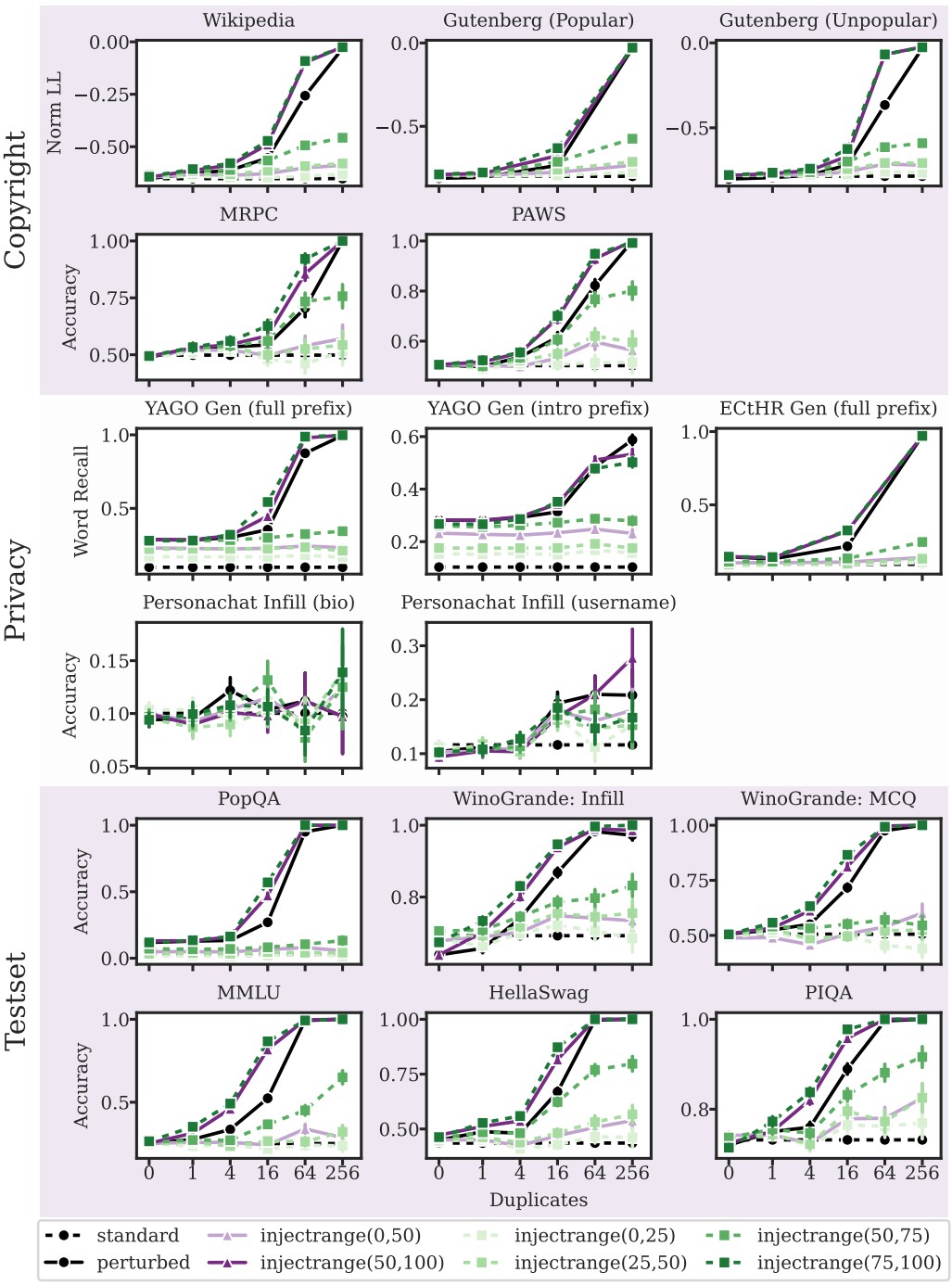

Figure 14: **Evaluation on the InsertRange models.** Models that were trained on perturbations only in the early stages of training have lower performance on the memorization tasks than models trained on perturbations in the late stages of training. InsertRange(x,y) denotes a model trained on a corpus with perturbations inserted in batches between x% and y% of training.

### E.2 PARAPHRASED RUNS

Two perturbed models (1B and 8B parameters) are trained on 100B tokens with the same perturbation data as the core perturbed model but with two data sets paraphrased: MMLU and YAGO Biographies. To prepare the data for the paraphrased runs, we construct paraphrased variants of the YAGO biographies and MMLU test set with `gpt-4.1-mini`. Unless otherwise noted, generation uses `temperature=1` and `top_p=1`. For each original perturbation example to be inserted, we obtain as many paraphrases as its required duplication count. The datasets are paraphrased as follows:

- **MMLU paraphrases.** We follow the paraphrasing instruction of Yang et al. (2023). When a paraphrase query is declined by `gpt-4.1-mini` API's safety filter, we use `gemini-2.5-flash-lite` with the same parameters.
- **YAGO paraphrases.** We adopt the diverse-style watermarking generation instructions from Cui et al. (2025). Each paraphrase is checked with a string-matching validator to ensure all biographical attributes are preserved. A paraphrase is accepted only if every attribute appears. We follow the procedure until we obtain the required number of valid paraphrases.

**PII can still be inferred from paraphrased biographies.** In Figure 15, the high accuracy of PII recontruction and inference indicates the paraphrase model has not just memorized a fixed string; instead, it generalizes to unseen queries for the PII, and this knowledge remains retrievable (similar to the retrievability observed in Allen-Zhu & Li, 2024). The accuracy of strong name-only attacks is higher on the 8B-parameter paraphrase model than on the original perturbed model at high duplication levels, indicating that models trained on paraphrases develop stronger semantic memory than the verbatim memory formed from training on exact duplicates. Personachat also shows the model's ability to retrieve memorized information in new contexts, and models can infer a user's persona based on the memorized chat logs (although the accuracy is low).

**PII can be leaked from paraphrased biographies with loss-based choice and generative evaluations.** The weakest attacks, which assume that the attacker has access to all PII about a person except one fact, are successful on models trained with paraphrased biographies. However, they have lower effectiveness than extracting the facts from the model that was trained on the original biographies. PII can be extracted with 100% accuracy from the core 8B perturbed model using the full prefix and full suffix MCQ format. This accuracy drops to 89% when extracting PII from the paraphrase model. Surprisingly, when using stronger attacks (attacker has access to only the persons name), PII is more accurately extractable from the 8B model trained on paraphrased biographies compared to the core models. However, this finding depends on the format of the attack and scale; generative evaluations cannot extract PII from the 1B paraphrased model.

**Models cannot generalize from paraphrased MMLU to the original examples.** We find that both models (1B and 8B parameters) obtain random accuracy on the MMLU MCQ evaluations when trained on paraphrased versions of the examples.

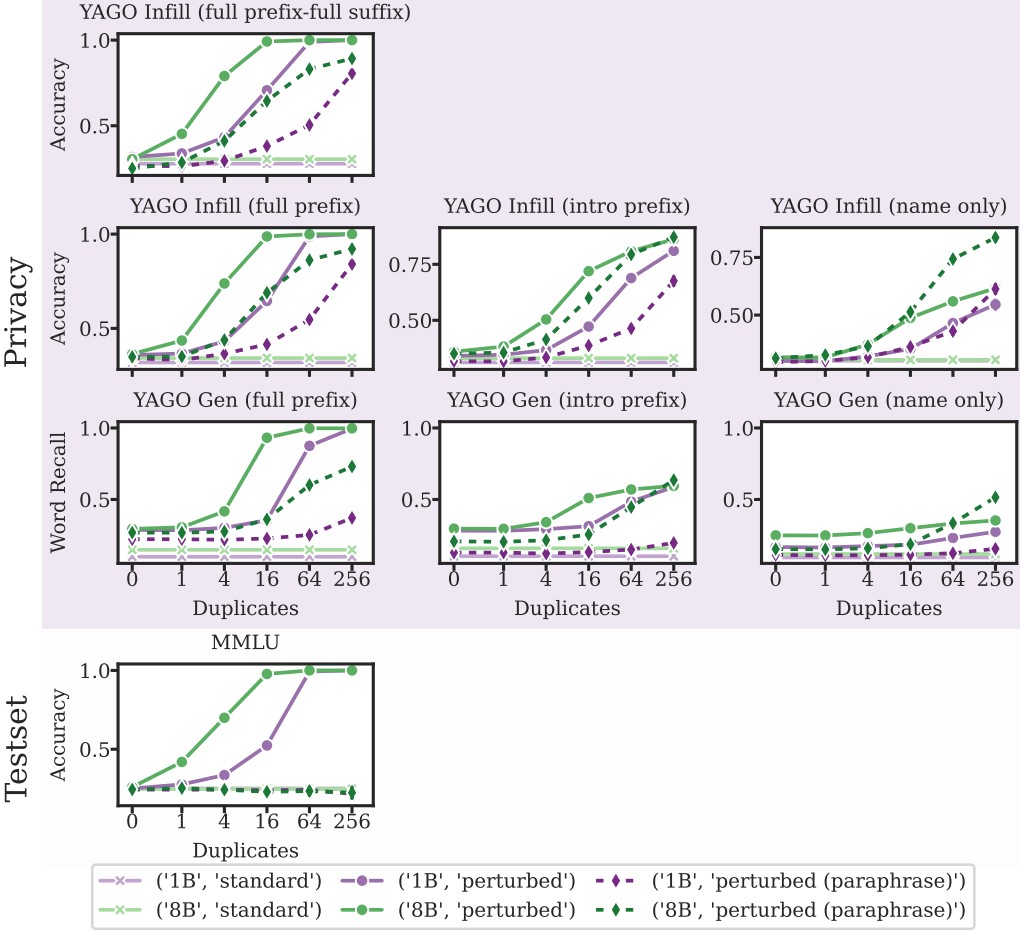

Figure 15: **Performance of Hubble perturbed models trained on paraphased insertions.** The models do not generalize from paraphrased examples seen in training to the original examples. However, PII can be reconstructed from models trained on paraphrased biographies, even with stronger attacks.

## E.3    ARCHITECTURE RUNS

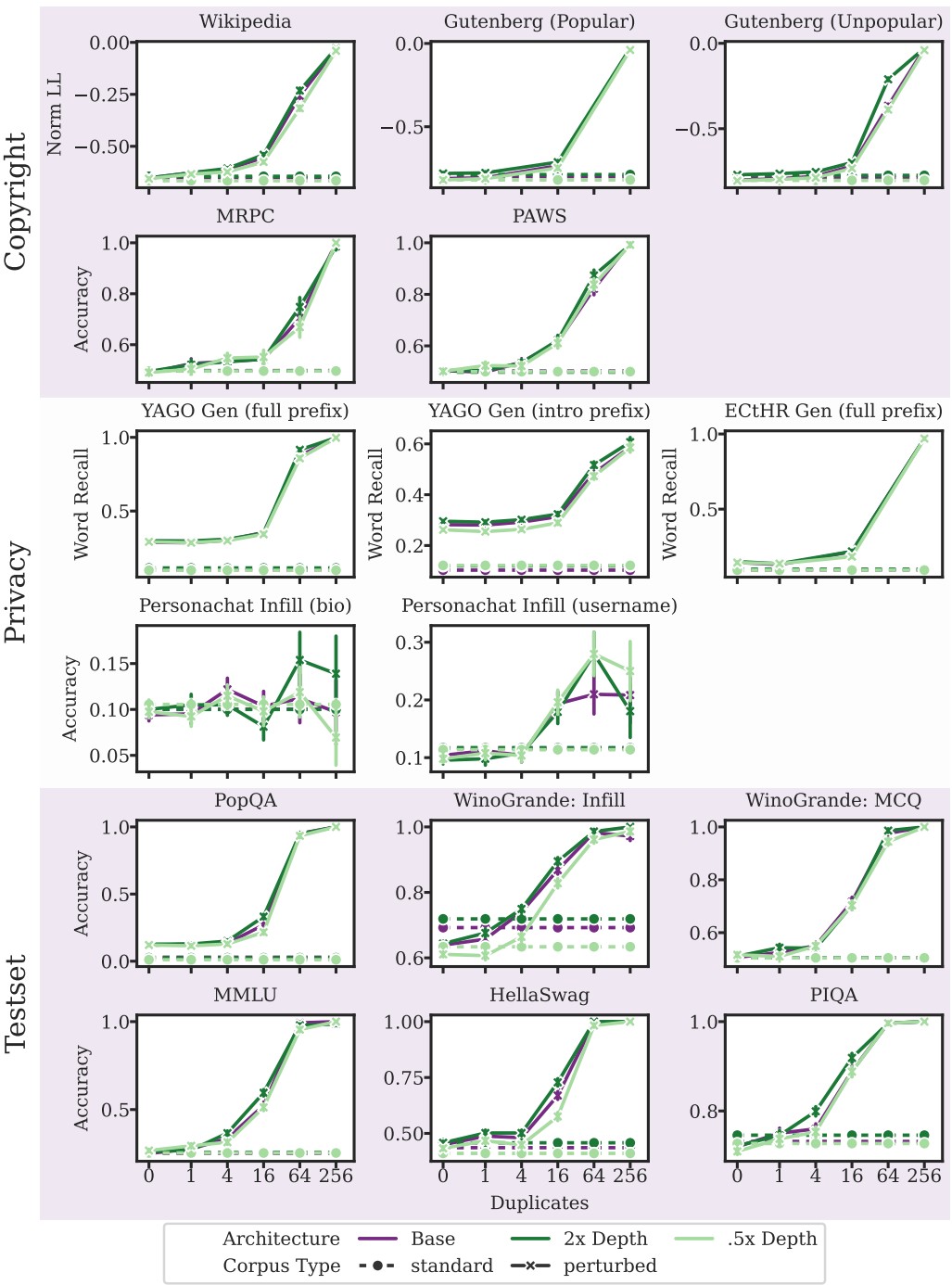

Figure 16: Deeper models memorize slightly more than shallower models. We train three 1B-parameter models with 8, 16, and 32 layers, adjusting width to keep total parameters constant ($\approx$ 1.2B). All models are pre-trained on 100B tokens. As shown in Figure 16, the deeper (narrower) model memorizes slightly more than the base 16-layer model, while the shallower (wider) model memorizes less.

## F    ADDITIONAL HUBBLEMIA RESULTS

We instantiate 12 variants of MIA benchmarks using the Hubble suite, using 4 models and 3 perturbation datasets (passages from Gutenberg Unpopular, biographies from YAGO, and contaminated examples from MMLU). As discussed in § 6.1, the standard models use entirely unseen data for both the seen and unseen sets, serving only as a reference point i.e. no method should achieve better-than-random accuracy in this setting.

- Table 11 reports MIA performance on the Hubble 8B Perturbed model.
- Table 13 reports MIA performance on the Hubble 8B Standard model.
- Table 12 reports MIA performance on the Hubble 1B Perturbed model.
- Table 14 reports MIA performance on the Hubble 1B Standard model.

Table 11: **ROC AUC scores of baseline MIAs for the HUBBLE 8B (500B tokens) perturbed model.** *Dup* indicates the duplication level of members. *Dup ≠ 0* treats all inserted perturbations as members. Non-members are always drawn from perturbations inserted 0 times. As duplication increases, memorization becomes stronger, and it becomes easier for membership inference attacks (MIA) to distinguish between members and non-members.

| Evaluation | MIA | HUBBLE 8B (500B tokens) Perturbed | | | | | |
|---|---|---|---|---|---|---|---|
| | | Dup ≠ 0 | Dup = 1 | Dup = 4 | Dup = 16 | Dup = 64 | Dup = 256 |
| Gutenberg Unpopular | Loss | 0.629 | 0.539 | 0.556 | 0.732 | **0.996** | **1.0** |
| | MinK% | 0.629 | 0.539 | 0.556 | 0.732 | **0.996** | **1.0** |
| | MinK%++ | **0.666** | **0.545** | **0.62** | **0.813** | 0.987 | 0.949 |
| | ZLib | 0.622 | 0.53 | 0.551 | 0.722 | **0.996** | **1.0** |
| Yago Biographies | Loss | 0.692 | 0.538 | 0.652 | **0.897** | **1.0** | **1.0** |
| | MinK% | 0.692 | 0.537 | 0.651 | 0.896 | **1.0** | **1.0** |
| | MinK%++ | **0.714** | **0.571** | **0.686** | 0.892 | 0.995 | 0.983 |
| | ZLib | 0.676 | 0.524 | 0.633 | 0.872 | **1.0** | **1.0** |
| MMLU | Loss | 0.673 | 0.529 | 0.628 | 0.857 | **1.0** | **1.0** |
| | MinK% | 0.672 | 0.529 | 0.626 | 0.854 | **1.0** | **1.0** |
| | MinK%++ | **0.743** | **0.58** | **0.731** | **0.943** | 0.994 | 0.986 |
| | ZLib | 0.644 | 0.523 | 0.593 | 0.775 | 0.993 | 0.999 |

Table 12: **Membership inference performance on various benchmarks with Hubble 1B Perturbed.** The Dup values indicate the composition of the seen set: for example, *Dup ≠ 0* means the attack compares all seen data against unseen data, whereas *Dup = K* means the attack compares unseen data against data that was included exactly $K$ times in the seen set.

| Evaluation | MIA | Hubble 1B Perturbed (500B tokens) | | | | | |
|---|---|---|---|---|---|---|---|
| | | Dup ≠ 0 | Dup = 1 | Dup = 4 | Dup = 16 | Dup = 64 | Dup = 256 |
| Gutenberg Unpopular | Loss | 0.552 | 0.52 | 0.504 | 0.552 | 0.73 | 0.999 |
| | MinK% | 0.552 | 0.52 | 0.504 | 0.552 | 0.729 | 0.999 |
| | MinK%++ | 0.575 | 0.513 | 0.53 | 0.605 | 0.825 | 1.0 |
| | ZLib | 0.543 | 0.511 | 0.497 | 0.533 | 0.729 | 1.0 |
| Yago Biographies | Loss | 0.606 | 0.506 | 0.557 | 0.696 | 0.928 | 1.0 |
| | MinK% | 0.606 | 0.506 | 0.556 | 0.695 | 0.927 | 1.0 |
| | MinK%++ | 0.615 | 0.509 | 0.565 | 0.715 | 0.947 | 1.0 |
| | ZLib | 0.596 | 0.499 | 0.551 | 0.679 | 0.899 | 1.0 |
| MMLU | Loss | 0.557 | 0.499 | 0.524 | 0.575 | 0.748 | 1.0 |
| | MinK% | 0.557 | 0.5 | 0.524 | 0.575 | 0.747 | 1.0 |
| | MinK%++ | 0.605 | 0.522 | 0.556 | 0.681 | 0.887 | 0.996 |
| | ZLib | 0.548 | 0.502 | 0.521 | 0.556 | 0.67 | 0.998 |

Table 13: **Membership inference performance on various benchmarks with Hubble 8B Standard**. The Dup values indicate the composition of the seen set: for example, *Dup ≠ 0* means the attack compares all seen data against unseen data, whereas *Dup = K* means the attack compares unseen data against data that was included exactly $K$ times in the seen set.

| Evaluation | MIA | Hubble 8B Standard (500B tokens) | | | | | |
|---|---|---|---|---|---|---|---|
| | | Dup ≠ 0 | Dup = 1 | Dup = 4 | Dup = 16 | Dup = 64 | Dup = 256 |
| Gutenberg Unpopular | Loss | 0.507 | 0.522 | 0.486 | 0.495 | 0.54 | 0.545 |
| | MinK% | 0.507 | 0.522 | 0.486 | 0.495 | 0.54 | 0.545 |
| | MinK%++ | 0.504 | 0.517 | 0.493 | 0.499 | 0.484 | 0.543 |
| | ZLib | 0.497 | 0.514 | 0.48 | 0.474 | 0.535 | 0.544 |
| Yago Biographies | Loss | 0.499 | 0.489 | 0.499 | 0.519 | 0.486 | 0.516 |
| | MinK% | 0.499 | 0.489 | 0.499 | 0.519 | 0.487 | 0.516 |
| | MinK%++ | 0.503 | 0.5 | 0.503 | 0.507 | 0.505 | 0.505 |
| | ZLib | 0.495 | 0.479 | 0.5 | 0.523 | 0.481 | 0.495 |
| MMLU | Loss | 0.502 | 0.506 | 0.503 | 0.512 | 0.459 | 0.476 |
| | MinK% | 0.502 | 0.506 | 0.503 | 0.512 | 0.458 | 0.476 |
| | MinK%++ | 0.506 | 0.51 | 0.505 | 0.514 | 0.497 | 0.45 |
| | ZLib | 0.501 | 0.505 | 0.504 | 0.506 | 0.463 | 0.495 |

Table 14: **Membership inference performance on various benchmarks with Hubble 1B Standard.** The Dup values indicate the composition of the seen set: for example, *Dup ≠ 0* means the attack compares all seen data against unseen data, whereas *Dup = K* means the attack compares unseen data against data that was included exactly $K$ times in the seen set.

| Evaluation | MIA | Hubble 1B Standard (500B tokens) | | | | | |
|---|---|---|---|---|---|---|---|
| | | Dup ≠ 0 | Dup = 1 | Dup = 4 | Dup = 16 | Dup = 64 | Dup = 256 |
| Gutenberg Unpopular | Loss | 0.503 | 0.517 | 0.484 | 0.494 | 0.534 | 0.531 |
| | MinK% | 0.502 | 0.517 | 0.483 | 0.494 | 0.534 | 0.531 |
| | MinK%++ | 0.5 | 0.509 | 0.493 | 0.497 | 0.481 | 0.529 |
| | ZLib | 0.493 | 0.509 | 0.477 | 0.471 | 0.529 | 0.533 |
| Yago Biographies | Loss | 0.495 | 0.488 | 0.494 | 0.51 | 0.494 | 0.509 |
| | MinK% | 0.495 | 0.487 | 0.494 | 0.51 | 0.494 | 0.508 |
| | MinK%++ | 0.5 | 0.499 | 0.501 | 0.494 | 0.518 | 0.497 |
| | ZLib | 0.494 | 0.481 | 0.498 | 0.516 | 0.489 | 0.49 |
| MMLU | Loss | 0.502 | 0.506 | 0.502 | 0.519 | 0.459 | 0.48 |
| | MinK% | 0.503 | 0.506 | 0.502 | 0.519 | 0.459 | 0.481 |
| | MinK%++ | 0.509 | 0.512 | 0.509 | 0.53 | 0.475 | 0.448 |
| | ZLib | 0.501 | 0.504 | 0.503 | 0.508 | 0.465 | 0.494 |

## F.1    FULL ROC PLOTS

We provide full ROC plots for baseline MIA attacks on the 8B parameter, 500B token model (Figures 17, 18, 19). Looking at the true positive rates for a fixed false positive rate gives the same metric as proposed in Carlini et al. (2022). A table of examples is given in Table 3. In general, MinK%++ performs best. Attacks often do not achieve high TPR at low FPR.

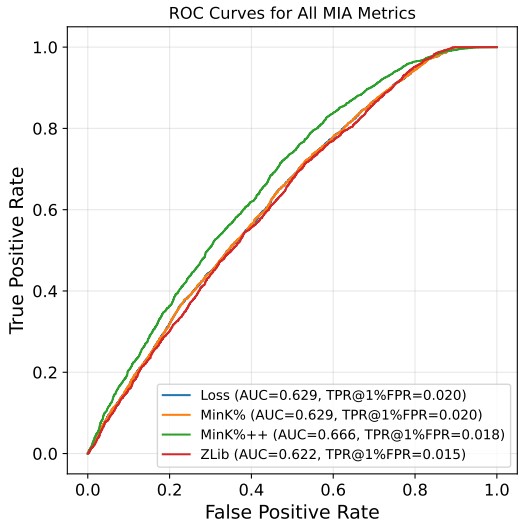

Figure 17: ROC plot for MIA attacks on Gutenberg Unpopular passages. Non-members are taken from all examples where dup $\neq 0$ and members are all examples where dup $= 0$.

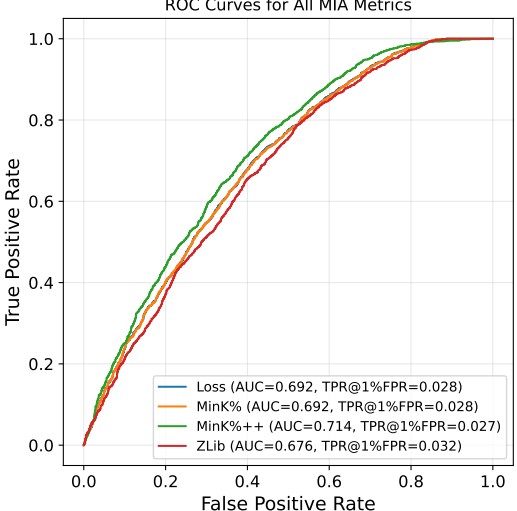

Figure 18: ROC plot for MIA attacks on YAGO biographies. Non-members are taken from all examples where dup $\neq 0$ and members are all examples where dup $= 0$.

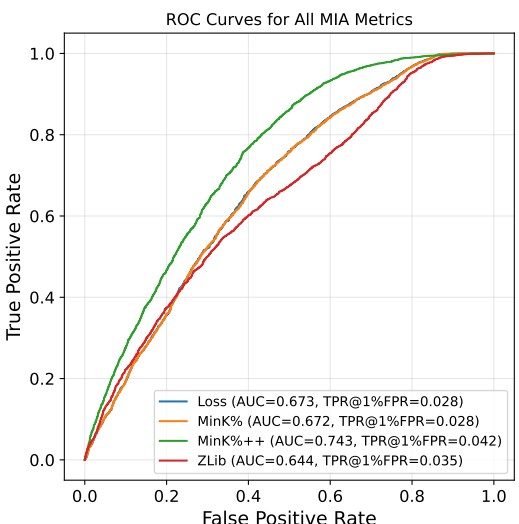

Figure 19: ROC plot for MIA attacks on MMLU examples. Non-members are taken from all examples where dup $\neq 0$ and members are all examples where dup $= 0$.

# G   ADDITIONAL HUBBLEUNLEARNING RESULTS

Below are the detailed hyperparameters for each method:

| Hyperparameter | RMU | RR | SatImp |
|---|---|---|---|
| **Training type** | Layer FT | LoRA FT | Full FT |
| **Layers / Targets** | 5, 6, 7 | 10, 20 (transform all) | — |
| **LoRA Rank / $\alpha$ / Dropout** | — | 16 / 16 / 0.05 | — |
| **LoRRA $\alpha$** | — | 10 | — |
| **Alpha ($\alpha$)** | 100, 1000, 10000 | — | 0.01, 0.1, 1 |
| **Steering coefficient** | 5, 50, 500 | — | — |
| $\beta_1$, $\beta_2$ | — | — | (5, 6), 1 |
| **Learning rate** | 5e-5, 1e-5, 5e-4 | 5e-5, 1e-4, 5e-4, 1e-3 | 1e-5, 5e-5, 1e-4 |
| **Effective batch size** | 4 | 8 | 16 |
| **Epochs** | 4, 8 | 4, 8 | — |
| **Sample max length** | 512 | 256 | 256 |

Table 15: **Grid search configurations for unlearning methods.** Each method is tuned over the listed hyperparameters. RMU and RR involve partial fine-tuning, while SatImp uses full fine-tuning.

We provide the full scale unlearning results for Gutenberg in Figure 20 and YAGO in Figure 21.

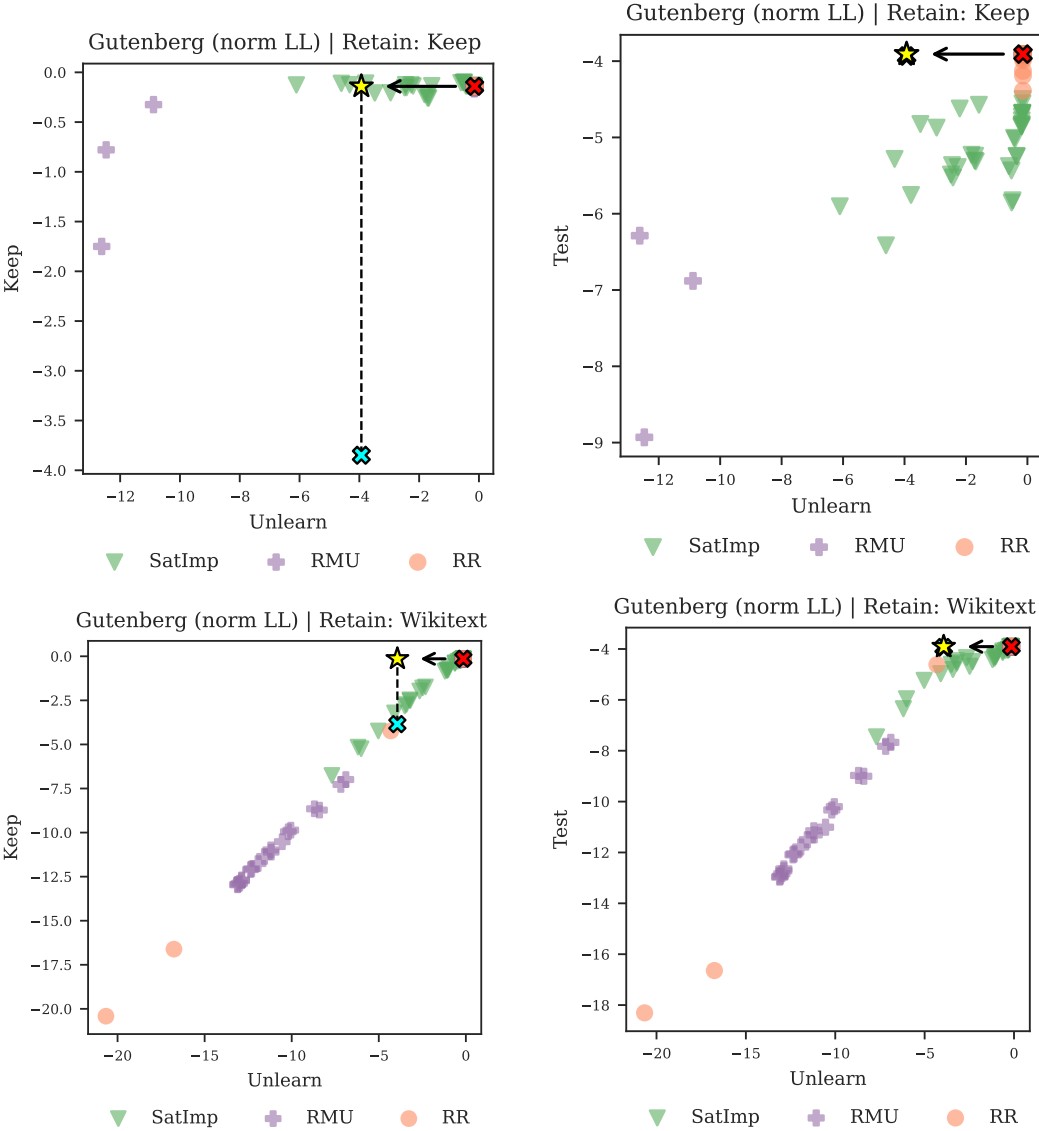

Figure 20: **Unlearning results on Gutenberg Unpopular.** Unlearning results using (out-of-domain, unseen) Wikitext (lower row) and (in-domain, seen) Keep set (upper row) as the retain sets. None of the unlearning methods simultaneously achieve the target behavior on both the seen Keep set (left column) and the unseen Test set (right column).

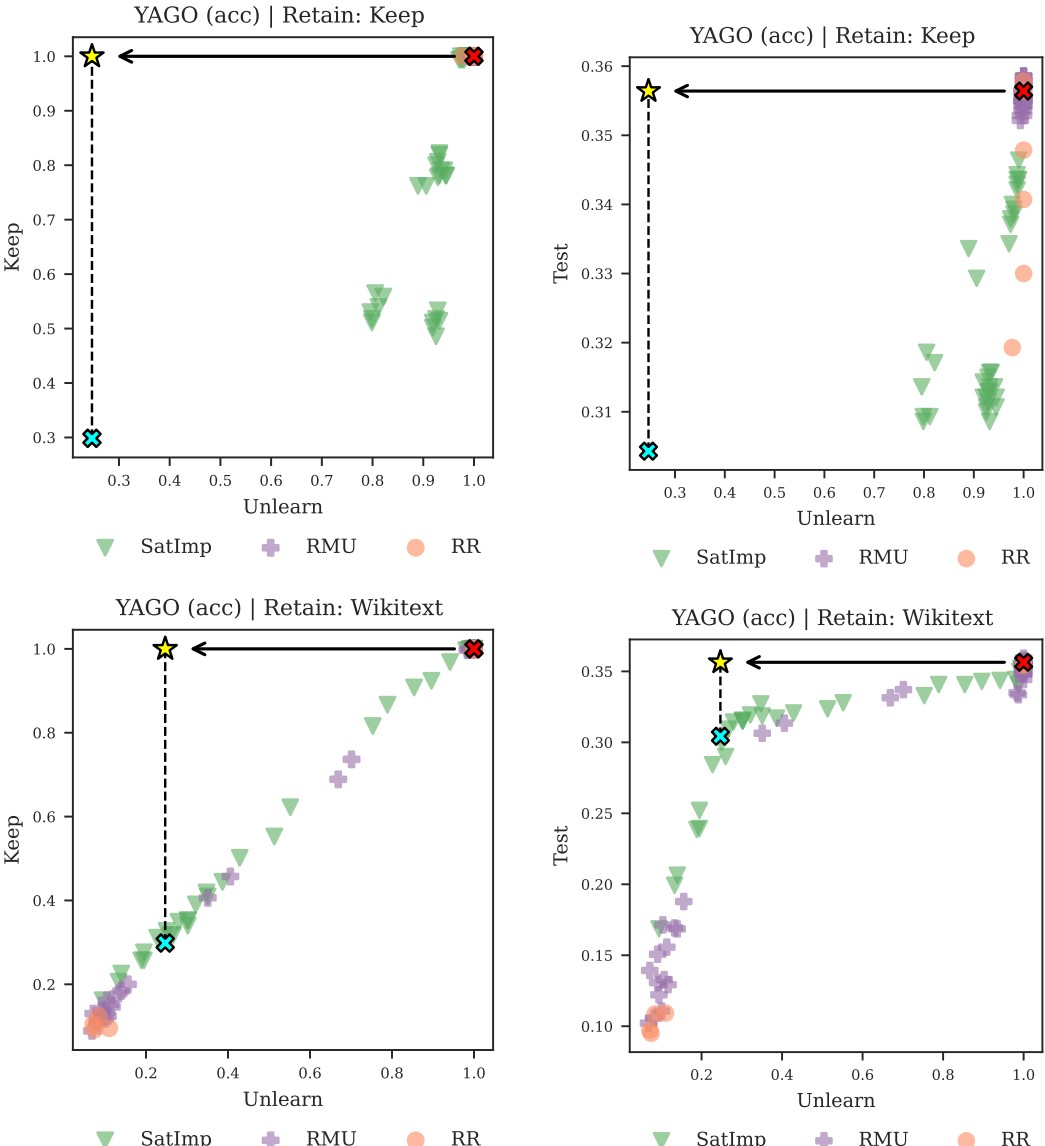

Figure 21: **Unlearning results on YAGO biographies.** Unlearning results using (out-of-domain, unseen) Wikitext (lower row) and (in-domain, seen) Keep set (upper row) as the retain sets. None of the unlearning methods simultaneously achieve the target behavior on both the seen Keep set (left column) and the unseen Test set (right column).

## H  FURTHER DISCUSSION

**How is information memorized?**  Understanding how transformers memorize is a basic scientific question that has been studied extensively in the literature (Geva et al., 2021; Dai et al., 2022, among others). A better understanding of the mechanisms of model memorization can inform the design of knowledge editing or unlearning techniques (Meng et al., 2022). Another practical application is in separating out knowledge from model parameters and enabling the responsible use of data (Shi et al., 2025a; Ghosal et al., 2025). With the perturbations in HUBBLE, interpretability studies can analyze a wide range of causal effects and control for factors such as the duplication rate or timing of an inserted text. The randomness in the perturbation data (e.g., the synthetic biographies) may also be useful as canaries to probe whether knowledge is localized to certain parameters (Maini et al., 2023; Chang et al., 2024b). Finally, the released checkpoints enable the study of how memorization evolves throughout training (Biderman et al., 2023a; Chang et al., 2024a).

**How can memorization be measured?**  For debates around copyright and privacy, there is a need for more intuitive and robust memorization metrics (Schwarzschild et al., 2024, as an example). HUBBLE perturbations span diverse data types that enable the development of new metrics, and the controlled insertions can validate these measurements (the same property that makes HUBBLE a solid benchmark for membership inference). Measuring memorization is closely related to privacy auditing, as both aim to detect whether a model reveals information about specific training examples; borrowing intuitions from differential privacy, such as bounding sensitivity, may be useful here (Panda et al., 2025). For a number of tasks within HUBBLE, model performance reflects a combination of both memorization and generalization (Feldman & Zhang, 2020), and isolating memorization effects may require advanced attribution methods (Ilyas et al., 2022; Grosse et al., 2023).

**How can memorization be mitigated?**  HUBBLE establishes two best practices—dilution and ordering—for mitigating memorization. HUBBLE's perturbation data is designed to emulate memorization risks across domains, and the models provide a testbed for evaluating new mitigation strategies. One direction to explore is whether quantization can generally reduce memorization risks as well (Chang et al., 2025; Kumar et al., 2025). Because memorization and data poisoning both rely on how models internalize specific examples, advances in mitigation may also reduce poisoning vulnerabilities; for instance, ordering has been found to influence the strength of poisoning attacks (Souly et al., 2025). Beyond identifying mitigation strategies, understanding their limitations is equally important. Best practices such as dilution may reduce memorization but may not fully eliminate all copyright or privacy concerns (Cooper et al., 2024; Mireshghallah & Li, 2025).

# I ADDITIONAL PLOTS

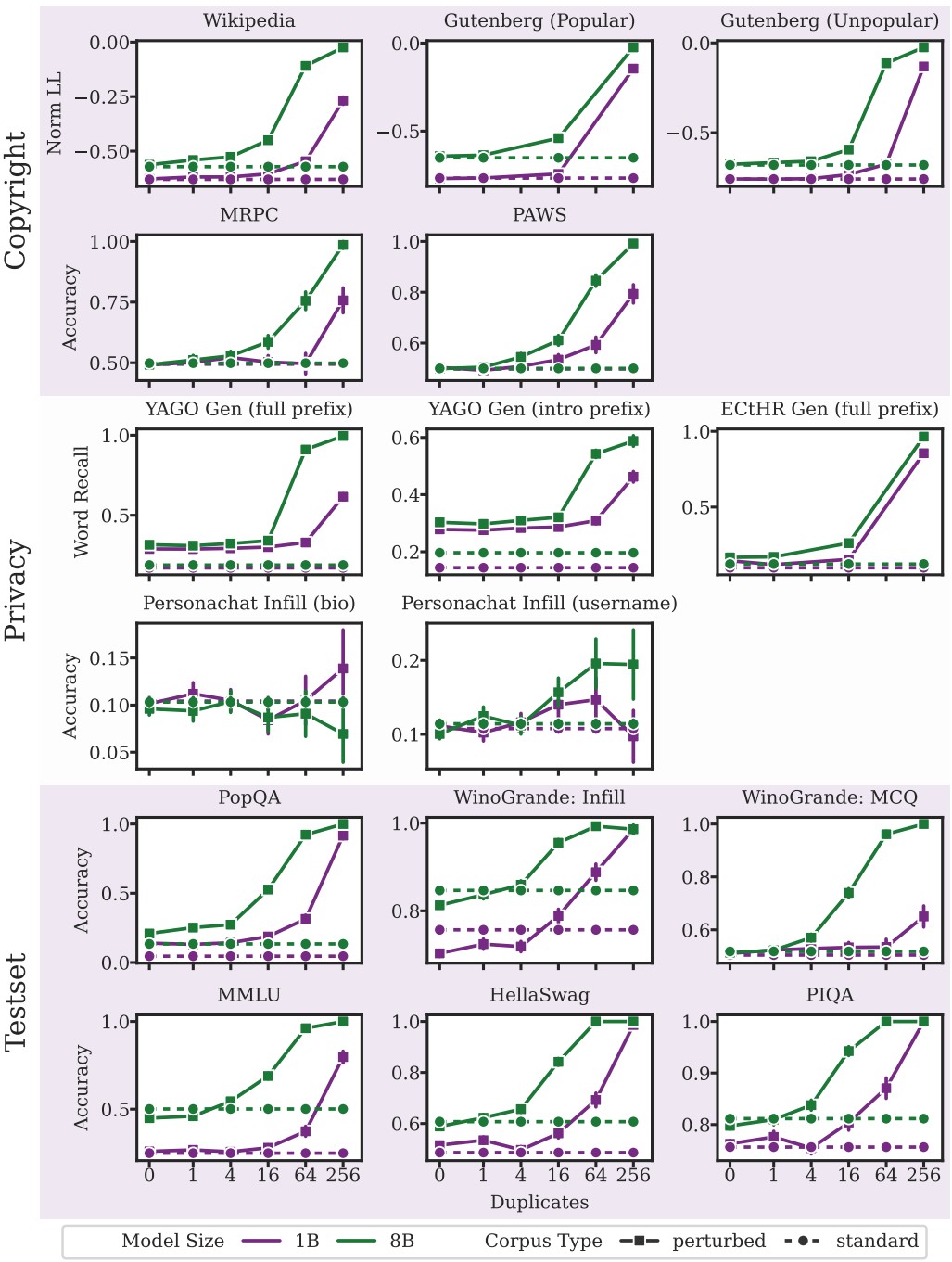

Figure 22: **Larger models memorize at lower duplicates.** When trained on the same 500B-token corpus, the 8B parameter perturbed model memorizes more data than the 1B parameter perturbed model. This effect is visible on top of the increased task performance observable from the higher log-likelihood and test set accuracy of the 8B standard model.

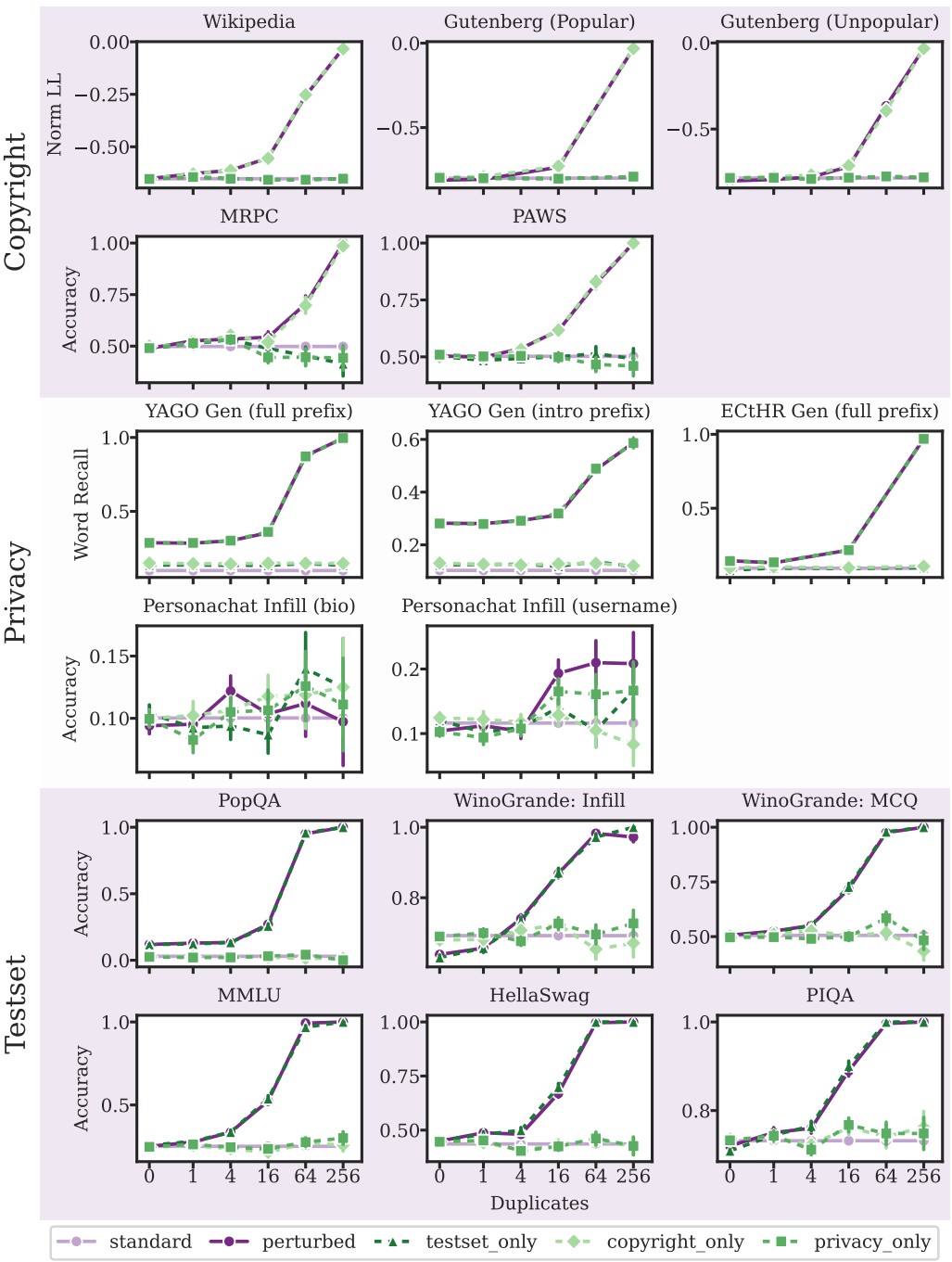

Figure 23: **The perturbed model matches the behavior of domain-specific models on the respective set of evaluations.** The perturbed model matches the copyright_only model in memorizing the copyright passages and paraphrases, privacy_only model in generating memorized PII from biographies and chat, and testset_only model in memorizing the testsets. Thus, the perturbed model can be used to study individual domains despite being jointly trained on all three domains.

