# OpenReview forum: "Hubble: a Model Suite to Advance the Study of LLM Memorization"
_ICLR.cc/2026/Conference — ICLR 2026 Oral_

### Official Review · Reviewer_vHGJ · 2025-10-17

**Soundness:** 4
**Presentation:** 4
**Contribution:** 3
**Rating:** 8
**Confidence:** 3

**Summary:**

This paper presents a set of open-source large language models (LLMs) called Hubble, which is designed for the scientific study of LLM memorization. The Hubble models spans 1 billion to 8 billion parameters and are trained on corpora of 100 billion to 500 billion tokens. Importantly, the models come in pairs: standard models trained on a large English corpus and perturbed models that include controlled insertions of sensitive text (like book passages, biographies, and test sets) to simulate memorization risks. During the training process, the authors conduct experiments to understand how the insertion of sensitive data affects memorization. Their analysis reveals that (1) diluting sensitive data by increasing the training corpus size can reduce memorization risks, (2) inserting sensitive data earlier in the training process can help mitigate these risks, (3) larger models memorize with lower duplications and (4) perturbations from different domains minimally interfere with each other. As for the use cases, the authors demonstrate that the Hubble models can be used to evaluate membership inference and machine unlearning methods, providing a controlled environment for further research in these areas.

**Strengths:**

- This paper is a fully open-source contribution to the field of LLM memorization, providing a suite of models that can be used for later research.
- The training methodology is faithfully described, and the analysis of the results is thorough and valid, without overclaiming as far as I can tell.
- The amount of experiments and analysis is impressive.

**Weaknesses:**

I am not fully convinced that the published models will add much value to the research community. In general, when people pick a model, they tend to pick commercial ones like LLaMA or Qwen, therefore, it is likely that the Hubble models will be used in special cases as discussed in Section 5. I am not an expert in MIA, so I cannot judge the value of the models in that context. However, I am not sure if what was discussed in Section 5.2 is a desirable benchmark for unlearning. The Huddle benchmark randomly separates the 256-duplicated set into a Unlearn set and a Keep set, meaning that these two sets follows the same distribution. As a result, we see in Figure 3 that unlearning methods fail to reduce the memorization of the Unlearnset without hurting the performance on the Keep set. However, in real unlearning tasks, the forget set and the retain set are usually from different distributions, and unlearning algorithms are usually *expected* to generalize the forget set and remove related but not identical information. From this perspective, the Hubble benchmark might be too strict compared to many unlearning tasks.

But overall, I think this is a solid paper with a good amount of experiments and analysis, and I recommend acceptance.

**Questions:**

Can you share your comments on the weaknesses above?

---

> ### Author Response · Authors · 2025-11-25
> **Thank you for your review!**
>
> Thank you for your time and feedback. We really appreciate your highlight on the amount of experiments and analysis. These analyses represent many months of work and this recognition means a lot to us.
>
> ### **W1. Will the published models add value to the research community?**
> We designed Hubble to broadly connect with the entire field of LLM memorization, and we hope that it will enable researchers to make important discoveries on memorization. For the purposes of studying memorization, it is important to study fully open models like ours, Pythia, or Olmo, where everything the model sees during training is known. Models like Llama and Qwen are only open weight and do not provide their training data. We also want to point out that Pythia is often the starting point for memorization research and has 1500+ citations. Our model has all the properties of the Pythia model suite but our models have newer architectures and are trained on slightly more data (500B compared to 300B).
>
> ### **W2. For unlearning, the unlearn and keep sets come from the same distribution.**
> Many types of unlearning benchmarks exist, and for safety related unlearning (e.g. unlearning how to make dangerous biological weapons) the distributions of the unlearn and keep sets typically differ (biological knowledge vs. general knowledge). Our setting is still interesting for privacy or copyright settings, where you might want to unlearn some books but keep others, and Hubble shows that this is difficult and requires precise unlearning.

---

> > ### Comment · Reviewer_vHGJ · 2025-11-25
> >
> > I would like to thank the authors for their rebuttal.
> >
> > **For W1.** I agree with the authors' point.
> >
> > **For W2.** I do think even if in privacy or copyright settings, the the unlearn and retain distributions still differ. I think we can agree to disagree on this subjective topic.
> >
> > Overall, this rebuttal does not change my opinion on this paper since I already recommended acceptance.

---

### Official Review · Reviewer_XuhG · 2025-10-20

**Soundness:** 3
**Presentation:** 3
**Contribution:** 4
**Rating:** 6
**Confidence:** 3

**Summary:**

This paper's main contributions are as follows:
1. Hubble, a suite of open-source pretrained LLMs for memorization research
2. A memorization study using those models
3. Benchmarks for membership inference and unlearning using Hubble models

**Hubble model suite**: The core Hubble suite consists of 8 models that span the combinatorial space of 1B vs. 8B parameters, 100B vs. 500B pretraining tokens, and with vs. without added "perturbations". Perturbations are documents that are injected into the base corpus (DataComp-LM); they are designed to allow studies of copyright, privacy, and benchmark contamination. Memorization can be studied by comparing pairs of models with and without added perturbations. The Hubble suite contains further "ablation models", measuring effects of data ordering, interference between data domains, paraphrasing, and architecture choices. All models use a modified Llama 3.X architecture and training is fully open-source (with known datasets and ordering).

**Memorization study**: The paper performs a broad memorization study on the Hubble models. Most notably, the study confirms that memorization increases with data duplication, that larger models memorize more, and that data appearing earlier in training is memorized less. Furthermore, training on a few samples of a benchmark's test set does not necessarily improve performance on that benchmark. The authors also verify that simultaneously including multiple data domains at once does not lead to interference effects between domains.

**Membership inference and unlearning benchmarks**: Lastly, the paper proposes to use the Hubble models as a benchmark for membership inference and unlearning. The authors perform a small case-study for both types of benchmarks by comparing a few representative methods. They find that none of the evaluated methods manage to unlearn target knowledge and simultaneously preserve capabilities on non-target data.

**Strengths:**

**Highly useful model suite**: The Hubble model suite is highly useful for memorization research, and hence serves as a more contemporary test bed compared to, say, Pythia. For one, the 8 core models allow studying the causal effects of adding certain types of data in a broad range of domains. The authors choose domains and datasets well, and they demonstrate that the models are useful to measure things beyond pure verbatim memorization.

The model ablations ("Runs" on p. 4) are also noteworthy. For one, they allow the authors to verify their setup (e.g., that there is no interference between domains; Figure 20). In addition, due to their design, I can see the model ablations being useful for future research (even beyond memorization).

**Broad memorization study and benchmarks**: The authors not only propose a useful model suite, but also perform a plethora of studies on those models. This includes confirmation of already established phenomena (e.g., data is memorized less if it is seen earlier in training), but also new insights (e.g., test set contamination does not necessarily improve benchmark scores, or that common unlearning methods fall short of their goal). Due to the breadth of the memorization study, and the additional benchmarks, I think those contributions are already significant in isolation.

**Polished and clear presentation**: Overall, the paper seems polished; it is mostly well-organized and easy to understand. The authors provide a lot of detail and transparency on their training and evaluation procedures. In the memorization study, findings that confirm existing observations/beliefs are contextualized with the relevant related work. I also like Figure 4; it intuitively illustrates how the base corpus is modified to include perturbations.

**Insights and tools for model training**: The insights from this paper could be useful beyond memorization research. The authors mention that their research resulted in additional tools that could benefit future research (Footnote 2 on page 4). Additionally, assuming the full training pipeline code is documented well, it might even help other researchers train similar suites of models.

**Weaknesses:**

While this is an overall high-quality paper, there is one potential major issue related to decontamination between the base training corpus and perturbations. I am happy to raise my score if those issues are addressed.

**Potential contamination between base corpus and perturbations**: The decontamination step of Hubble's data pipeline focuses on high precision (i.e., only discarding samples that clearly overlap the base corpus and perturbations). However, false negatives (i.e., contamination that is not detected) are much more harmful for memorization research, as they can taint the results for the use cases in this paper (studying memorization, MIA/unlearning benchmarking).
For one, the authors verify matches manually; however, this can only detect false positives and not false negatives. Moreover, decontamination for perturbations (of non-trivial length) only uses exact matches of at least 20 tokens. Hence, it is possible that non-trivial overlaps between perturbations and base corpus exist (e.g., with slightly different formatting).
I thus argue that, to ensure results of memorization studies are valid, this requires further discussion and a potential empirical study of shorter overlaps between base corpus and perturbation sequences.

**Membership inference benchmark is incomplete**: The membership inference case-study (Section 5.1 and Appendix F) exclusively uses the ROC AUC score as a metric. This is well-known to be misleading and obfuscating important details [(Carlini et al., 2021)](https://arxiv.org/abs/2112.03570). Instead, I recommend reporting the TPR at a low fixed FPR (e.g., 1%), and providing full ROC plots.
Additionally, I could not find the size of the member and non-member sets mentioned. However, this information is crucial to assess the resolution of the reported metrics (small set sizes might yield high variance in reported scores).

**Relatively short pretraining compared to "real-world" models**: Hubble is only trained on 0.5T tokens, compared to 4T and 9-15T tokens for OLMo and Llama 3.X models of similar sizes. The authors are transparent about this. Yet, the insertion timing results have to be viewed relative, and Hubble might not allow studying memorization due to emergent behaviors from very long pretraining. On the flip side, while Hubble models are obviously not SotA on standard benchmarks, they seem to at least often perform on-par with similarly-sized Llama 3.1/3.2 (but not OLMo 2) models. I understand that this limitation is unavoidable due to the large cost of pretraining.

**Minor feedback on figures**:
1. Most plots only use a single hue for all lines and distinguish them by brightness (see, e.g., Figures 1 and 2). I think the plots would be easier to read and compare if they used very different colors, especially in plots with many lines.
2. I found it hard to process Figure 1, particularly since it contains many small plots for many settings. At that point in the paper, I would find it easier to have a high-level figure that highlights an aggregate or a subset of results, and defer the full Figure 1 to the appendix.
3. In contrast, Section 4.1 provides a lot of interesting insights, but defers most plots to the appendix. The paper could be easier to read if the main matter showed (a subset of) the results discussed there (to avoid jumping between sections) and deferred detailed plots (such as Figure 1) to the appendix.

**Questions:**

1. How are duplication counts sampled? Is it completely uniformly at random, or are the numbers balanced (beyond using 256 duplicates less often)? From Footnote 3 on p. 4, I believe it to be the former, but I could not find an exact statement.
2. What exactly are the held-out perturbations (e.g., perturbations with duplicate count 0 in Section 5.1)? Will those be published with the results of this paper in an easily-accessible format? I believe that this is an crucial point for benchmarking membership inference; other researchers might theoretically be able to generate such a held-out set themselves, but this requires non-trivial engineering effort that the authors of Hubble already did.
3. Similarly, what are the sizes of the member and non-member sets in the membership inference benchmarks?
4. How difficult would it be to extend Hubble to post-training (e.g., instruction tuning)? This might be a highly useful extension, given that there are even fewer benchmarks to measure memorization in post-training.
5. From my understanding, different repetitions and number of training tokens change the relative _frequency_ of perturbations. This terminology is mentioned consistently throughout the paper, except for the beginning of Section 4. That part of the paper also mentions the _spacing_ between perturbations. Is the spacing something that is effectively controlled? Or does the beginning of Section 4 just assume that spacing and frequency are negatively correlated? In the latter case, I would either only mention frequency or empirically validate that lower frequency indeed yields uniformly larger spacing between perturbations.

---

> ### Author Response · Authors · 2025-11-25
> **Thank you for your review!**
>
> Thank you for your time and feedback. This review is extremely thorough, and we really appreciate you highlighting so many of our contributions. These points represent many months of work, and the recognition means a lot to us.
>
> ### **W1. Potential contamination between base corpus and perturbations.**
> This is an important point and one which we took seriously during preprocessing. On decontamination, we want to first point out that capturing all instances of contamination in pretraining is extremely difficult. Using a high, 20-gram threshold to filter out exact matches was a deliberate design decision. If we had set it too low, we would have removed many non-duplicate texts which are similar to our perturbations. As n-gram overlap statistics likely affect how the perturbations are memorized, choosing a lower threshold runs the risk of distorting the n-gram overlap statistics for our perturbations. On the other hand, many of our perturbations were also carefully constructed to minimize the possibility of contamination. For instance, our biographies are synthetic and randomly generated and do not appear in the rest of the pretraining corpus. The Wikipedia passages were exclusively taken from articles created after the cutoff date of DCLM. For test sets, we also insert new test sets (Ellie and Munch) released after the DCLM cutoff date.
>
>
> ### **W2. Membership inference benchmark is incomplete.**
> ROC AUC score is misleading. This is a good point and we acknowledge that reporting TPR at a low fixed FPR is more relevant from a security/privacy perspective. We have provided a few ROC plots in Appendix G.3. Given that the members and non-members are identically sampled and randomly partitioned, using ROC AUC is still sound and intuitive, and can make sense from a perspective of meausuring memorization.
> We added Table 2 in the appendix which tallies how many examples are duplicated at each level, for each dataset.
>
> Minor feedback: We thank the review for their diligent feedback on our presentation. We will try to incorporate these suggestions for the final version.
>
> ### **Q1. How are duplication counts sampled?**
> We pick bin sizes based on a basic rule of thumb: we aimed to include at least 1000 examples in the “1” bin (for binary tasks, this is often enough to get reasonably wide error bars) and then as the duplicate counts grow larger, we reduce the number of examples in that bin to reduce the amount of duplicated text (usually including as many “0” examples as non-zero duplicated examples, if possible). For the “0” bin, we included a large number of examples as they are not duplicated in the model. Figure 1 shows that the error bars in the evaluation are reasonable.
>
> ### **Q2. What exactly are the held-out perturbations?**
> We first generated all the perturbations, and then randomly assigned them to be duplicated 0 or more times. This means that the non-members are sampled identically to the members. And of course, all our datasets will be accessible on Hugging Face. These datasets will include the held out perturbations, which is useful for many types of analyses.
>
> ### **Q3. How difficult would it be to extend Hubble to post-training (e.g., instruction tuning)?**
> This is definitely possible, and the 8B model size can be finetuned on many consumer GPUs. We leave this to future work.
>
> ### **Q4. Is the spacing something that is effectively controlled?**
> No, we don't directly control the spacing and, like you mention, assume that relative frequency is inversely correlated with spacing. Since the perturbations are randomly scattered across training, as the duplicates increase, spacing between examples should decrease. We thought this would be a useful intuition, but the relative frequency wording is more accurate. We have updated this text in the paper.

---

> > ### Comment · Reviewer_XuhG · 2025-11-25
> > **Reply to rebuttal**
> >
> > I thank the authors for their explanations and clarifications that resolve all my questions.
> >
> > On **W1.**: I thank the authors for the discussion. It convinced me that the choice of 20-gram threshold is reasonable and that the amount of potential contamination is reasonably low.
> >
> > On **W2.**: I appreciate the ROC plots in Appendix G.3 as they provide a much more informative picture. However, I would like to ask authors to change them into log-log plots, as is typical for reasons discussed in [Carlini et al., 2021](https://arxiv.org/abs/2112.03570). Additionally, I have to disagree with the following statement:
> >
> > > Given that the members and non-members are identically sampled and randomly partitioned, using ROC AUC is still sound and intuitive, and can make sense from a perspective of meausuring memorization.
> >
> > While the *data* might be uniformly distributed, *memorization/privacy risk* is typically not (see, e.g., [Carlini et al., 2021](https://arxiv.org/abs/2112.03570), [Carlini et al., 2022](https://arxiv.org/abs/2206.10469), [Aerni et al., 2024](https://arxiv.org/abs/2404.17399)), hence I believe the "right" thing to report in Table 1 is TPR at (say) 1% FPR. Given that it is just a different way of aggregating membership scores, I also think that this should be a quick change.
> >
> > Nevertheless, this is a small detail, and my major concern about contamination has been resolved. I hence increased my score.

---

### Official Review · Reviewer_HMNX · 2025-10-31

**Soundness:** 4
**Presentation:** 4
**Contribution:** 3
**Rating:** 8
**Confidence:** 4

**Summary:**

This work introduces a suite of (not yet) open sourced models specifically for study of memorization in LLMs. HUBBLE consists of standard and perturbed model pairs trained on the same data with the latter containing controlled insertions of synthetic or real-world sensitive data.

On standard NLP benchmarks, HUBBLE models perform comparably to other open-weight models (e.g., Pythia) at similar scale, confirming training stability.

The authors train 8 primary models (1B and 8B parameters, 100B and 500B tokens) and 6 auxiliary models with variations in insertion timing, frequency, and architecture. Experiments on these trained models show two primary trends:

Dilution effect: memorization risk decreases when sensitive data are diluted within larger corpora.
Ordering effect: placing sensitive data early in pretraining reduces long-term memorization.
Model size effect: Larger models (8B) memorize faster and at lower duplication counts, aligning with previous scaling trends.
Minimal interference: Perturbations from distinct domains (privacy, copyright, contamination) do not interfere significantly.

This work also opens up the possibility for controlled benchmarks for MIA attacks and unlearning

**Strengths:**

The strengths of this work are listed as follows:

- This work would help form any following work overcome one of the primary drawbacks in this field, where experiments had to be either painfully small scale or had to rely on somewhat unknown data patterns of models trained without a focus on memorization studies
- The predictable and standard nature of the perturbations added to the training data would help any further study conduct clear A/B tests on this study
- I really like the detailed nature of the experiments especially the domain specific results

**Weaknesses:**

- This is an (computationally)expensive study and ideally more patterns could/should have been embedded into the training data to make the most out of it. One interesting direction might've been understanding the relation between the distance between paraphrased samples and the degree to which they're memorized

- Test set insertions focus on benchmarks like MMLU and HellaSwag, whereas contamination in real corpora can be indirect or paraphrastic.

- The work isolates memorization metrics from general model utility. While HUBBLE provides general evaluation benchmarks, it does not deeply analyze how memorization mitigation (via dilution or ordering) affects downstream performance or factual recall.

**Questions:**

- Please detail if there are a set of directions which could be explored using these models apart from the standard memorization and unlearning tasks for which this study is directly intended for

---

> ### Author Response · Authors · 2025-11-25
> **Thank you for your review!**
>
> Thank you for your time and feedback. We really appreciate that you found our models and experimental design useful for future studies. We also thank you for noting the detailed nature of our experiments.
>
> ### **W1. This is an (computationally) expensive study and ideally more patterns could/should have been embedded.**
> Hubble contains 16 inserted datasets spanning 3 domains of copyright, privacy, and test set contamination. For several types of insertions, such as the YAGO biographies, we designed the task, generated the data, and implemented the corresponding memorization evaluation. We believe we accomplished a lot, but it is possible that we could have done more. As you mentioned, producing just these models is already computationally expensive, and we also want to point out that our work is done under time constraint as we had a fixed window in which we could use all our (gifted) compute. Your proposed experiment is interesting, and with the checkpoints you can already study this to some extent and measure how distance between insertions affects memorization.
>
> ### **W2. Contamination in real corpora can be indirect or paraphrastic.**
> We also released a paraphrased run, which was trained on paraphrased biographies and test sets (paraphrasing details in Appendix A.5 and results in Appendix E.2). This model can be used to study how paraphrasing affects memorization, and we find that training on exact duplicates generally causes stronger memorization.
>
> ### **W3. While HUBBLE provides general evaluation benchmarks, it does not deeply analyze how memorization mitigation (via dilution or ordering) affects downstream performance or factual recall.**
> In Appendix C.2, Table 4, we added general evaluations comparing 100B and 500B variants of Hubble perturbed. These 500B models have the dilution effect and generally slightly better performance. We also provide evaluations of ordering models and find that they are comparable to the core Hubble 1B 100B model.
>
> ### **Q1. Please detail if there are a set of directions which could be explored using these models.**
> The perturbations in Hubble were designed to broadly connect with the memorization literature. Besides the unlearning and memorization results presented in the paper, we hope that our models help make progress on several research directions:
> 1) Understanding how models memorize information using interpretability methods.
> 2) Studying when models generalize vs. memorize using the inserted test sets.
> 3) Designing better metrics to measure memorization for e.g. legal purposes.
> 4) Establish further best practices for mitigating memorization using Hubble as a testbed.

---

> > ### Comment · Reviewer_HMNX · 2025-11-26
> >
> > I thank the authors for the clarification and would be looking forward to seeing great work being built on top of this!

---

### Official Review · Reviewer_m7QY · 2025-11-01

**Soundness:** 4
**Presentation:** 4
**Contribution:** 3
**Rating:** 8
**Confidence:** 5

**Summary:**

This paper proposes HUBBLE, a comprehensive, open-sourced suite of LLMs and data to explore the memorization of pretrained LLMs. More specifically, it consists of 8 pairs of standard models and perturbed models with 1B or 8B and pretrained on 100B or 500B tokens. The authors conducted a series of experiments on the HUBBLE, which show conclusions consistent with the existing studies on LLM memorization and also provide new insights on how to mitigate the memorization of sensitive data. Moreover, the experimental results also show the effectiveness of HUBBLE to build benchmarks for use cases of the membership inference attacks and machine unlearning.

**Strengths:**

The strengths of the paper are listed as follows.
1. HUBBLE will be fully open-sourced upon publication, which is important for the researchers to reproduce its results and leverage the suite for their own study in LLM memorization.
2. The design of HUBBLE is reasonable and useful. The pair of standard models and perturbed models enables a fair and clean comparison when studying the mechanism of LLM memorization.
3. The open-source data and checkpoints also make the quantitative analysis of LLM memorization feasible. The unavailability of checkpoints or training data is a big bottleneck for the study of LLM memorization.
4. HUBBLE provides an insightful categorization of the potential risk caused by LLM memorization, which reflects the deep understanding of this research area and the practical safety demands in real-world applications.
5. The evaluation is comprehensive and solid. The numerical results, observations and conclusions drawn from HUBBLE are reasonable, convincing, and also consistent with the existing works in LLM memorization.

**Weaknesses:**

The major concern of HUBBLE is that it only covers the LLMs with 1B and 8B. Therefore, some observations from HUBBLE may not be extended to large LLMs, which usually have a higher memorization rate and are more widely adopted by commercial applications. However, for the academic research of LLM memorization, the two model sizes are sufficient.

**Questions:**

The questions are listed as follows.
1. In Figure 1, do the authors have any explanation for why dilution does not work well on the data samples duplicated 256 times?
2. For the experiments on the impact of the ordering of sensitive data in the pretraining, is the total number of sensitive data and their duplication times kept the same for all settings? For the setting with perturbed data across all four quarters, a specific sensitive data may be used in all four stages. However, for the setting with perturbed data only in the first quarter, will the same sensitive data be repeatedly used four times for fair comparison?

---

> ### Author Response · Authors · 2025-11-25
> **Thank you for your review!**
>
> Thank you for your time and feedback. We really appreciate that you find our survey of the literature insightful and our models, data, and evaluations useful.
>
> ### **Q1. Why does dilution not work well on the data samples duplicated 256 times?**
> In Figure 1, more duplicates causes stronger memorization, and we interpret dilution as left-shifting the memorization curve. That is, for the same number of duplicates the memorization will be weaker. At 256 duplicates, the memorization is very strong and even with dilution the memorization continues to be saturated (many metrics such as log likelihood are maxed out at the top of the plot).
>
> ### **Q2. Clarification on ordering experiments.**
> All the core and ordering models use the exact same set of perturbations. For the core models, all perturbations are randomly inserted across training. For the ordering models, all perturbations are randomly inserted into a quarter of training, so the perturbations are roughly 4 times as dense within that quarter, compared to core models. We set it up this way so the ordering models can be directly compared against the core models.

---

### Author Response · Authors · 2025-12-03
**Summary of rebuttal**

We thank all the reviewers for their feedback and their positive assessment of our work. To summarize for the AC, our initial scores were positive and our scores increased to (8,8,8,8) after rebuttal. Our highlights of the rebuttal are below:
* On potential contamination between perturbations and the training data (reviewer XuhG): after rebuttal, they noted our decontamination was reasonable and the chance of contamination was low, and increased their score.
* On whether the models will add value to the research community (reviewer vHGJ): after rebuttal, they agreed that the models are useful for memorization research due to the fully open-source distinction.

Again, thank you to all the reviewers for reviewing and participating in our rebuttal. This was an unusual cycle with many complaints, but our experience was quite positive. Thanks!

---

### Meta-Review · Area_Chair_DevF · 2026-01-04

**Summary:**

The paper presents Hubble, a series of LLMs for studying memorization. All reviewers recognize its contribution to the community, for example, by providing researchers studying memorization with alternative model choices beyond Pythia or GPT-Neo. Hubble has the potential to serve as a valuable testbed for research on memorization, machine unlearning, membership inference attacks, and related topics. During the rebuttal, the authors addressed concerns regarding potential contamination between perturbations and the training data, as well as the value of the Hubble models.

After carefully reviewing the paper, the reviews, and the rebuttal, the Area Chair agrees with the reviewers that Hubble is a valuable and timely contribution to research on LLM memorization and therefore recommends acceptance for an oral presentation.

**Reviewer Concerns:**

In the rebuttal, the authors addressed concerns regarding potential contamination between perturbations and training data, the value of the Hubble model suite, evaluation metrics for membership inference attack, and other experimental design details.

There remain a few questions for which reviewers and authors do not fully agree, such as the experimental setting for machine unlearning. These are better viewed as open research questions, and reviewers generally agree to disagree on these points.

**Reviewer Scores:**

Reviewers m7QY, HMNX, and vHGJ are expected to maintain their scores of 8 in support of acceptance. Reviewer XuhG indicated in their comments that they would increase their score to 8.

---

### Decision · Program_Chairs · 2026-01-26

Accept (Oral)